# Single-atomic platinum on fullerene $C_{60}$ surfaces for accelerated alkaline hydrogen evolution

Ruiling Zhang[1,4], Yaozhou Li[1,4], Xuan Zhou[2], Ao Yu[1], Qi Huang[1], Tingting Xu[1], Longtao Zhu[1], Ping Peng[1] ✉, Shuyan Song [2] ✉, Luis Echegoyen[3] ✉ & Fang-Fang Li [1] ✉

The electrocatalytic hydrogen evolution reaction (HER) is one of the most studied and promising processes for hydrogen fuel generation. Single-atom catalysts have been shown to exhibit ultra-high HER catalytic activity, but the harsh preparation conditions and the low single-atom loading hinder their practical applications. Furthermore, promoting hydrogen evolution reaction kinetics, especially in alkaline electrolytes, remains as an important challenge. Herein, Pt/$C_{60}$ catalysts with high-loading, high-dispersion single-atomic platinum anchored on $C_{60}$ are achieved through a room-temperature synthetic strategy. Pt/$C_{60}$-2 exhibits high HER catalytic performance with a low overpotential ($\eta_{10}$) of 25 mV at 10 mA cm$^{-2}$. Density functional theory calculations reveal that the Pt-$C_{60}$ polymeric structures in Pt/$C_{60}$-2 favors water adsorption, and the shell-like charge redistribution around the Pt-bonding region induced by the curved surfaces of two adjacent $C_{60}$ facilitates the desorption of hydrogen, thus favoring fast reaction kinetics for hydrogen evolution.

Hydrogen fuel is one of the most promising energy sources for the 21st century due to its cleanliness to the environment and high energy density. Among the current hydrogen production methods, electrocatalytic hydrogen evolution reaction (HER) is a promising technique for its simplicity and environmental friendliness[1,2]. However, HER kinetics, especially in alkaline electrolytes, is sluggish and complex due to a high energy barrier for the initial water adsorption and dissociation, and subsequent adsorption of *H and desorption of $H_2$ generated on the catalyst surfaces[3]. It is thus important to explore highly efficient alkaline HER catalysts.

Platinum (Pt) is known as the benchmark catalyst for HER, exhibiting a very low overpotential and fast kinetics due to its favorable hydrogen adsorption energy[4,5]. However, only the Pt surfaces intimately contacted with the electrolyte can participate in the catalytic reactions, which limits the full utilization of the Pt present and limits the HER activity. Hence, reducing the size of bulk Pt or Pt nanoparticles to single atoms can achieve nearly 100% Pt utilization in catalytic reactions[6–8]. Remarkably, Pt single-atom catalysts (SACs) have been successfully synthesized via wet-chemistry strategies[9,10], atomic layer deposition[11,12], high-temperature atom trapping from bulk particles[13,14], MOF-derived methods[15], and vacancies/defects immobilized strategies[16,17], and exhibit reasonably high catalytic performances. However, the current synthetic methods usually suffer from several issues associated with high temperature, complex synthesis process, and low single-atom loading (typically limited to one atomic percent (at%) or a few weight percent (wt%), significantly lower than that of the benchmark Pt/C (20 wt%), thus resulting in relatively limited catalytic activity and impeding their practical applications for hydrogen production[18]. It is worth noting that pursuing high-loading single atoms, especially at high-temperatures, results in undesirable

[1]State Key Laboratory of Materials Processing and Die & Mould Technology, School of Materials Science and Engineering, Huazhong University of Science and Technology, 1037 Luoyu Road, Wuhan 430074, P. R. China. [2]State Key Laboratory of Rare Earth Resource Utilization, Changchun Institute of Applied Chemistry, Chinese Academy of Sciences, 5625 Renmin Street, Changchun 130022, P. R. China. [3]Department of Chemistry and Biochemistry, University of Texas at El Paso, 500 West University Avenue, El Paso, TX 79968, USA. [4]These authors contributed equally: Ruiling Zhang, Yaozhou Li. ✉e-mail: ppeng@hust.edu.cn; songsy@ciac.ac.cn; echegoyen@utep.edu; ffli@hust.edu.cn

aggregation. Therefore, low-temperature processes for high-loading, high-dispersion Pt SACs are desired, but remain as a serious challenge.

On the other hand, the nature of the coordination of Pt is crucial for the catalytic performance as the coordination controls the charge distribution of the central Pt atom, allowing more modest binding of Pt to the $H_2O^*$, $H^*$, and $OH^*$ intermediates produced during the HER process, thereby enhancing the catalytic activity of Pt single-atom catalysts[19]. In addition, strong metal-support interactions (SMSI) can firmly anchor the metal single atoms, inducing effective electron transfer between the support and the supported metals. The most commonly studied supports to date are two-dimensional planar structural materials such as graphene[20], carbon nanosheets[21], transition metal chalcogenide[22,23], and layered double hydroxides (LDHs)[24]. Researchers recently discovered that a curved support matrix enables electron density accumulation around the single atom, thereby accelerating HER kinetics[25]. However, the loading of atomic Pt reported in this work was still very low at 0.27 wt%, and the catalytic activity needs to be improved.

Fullerenes are zero-dimensional nanocarbons with well-defined molecular structures that can be functionalized by chemical reactions at room temperature. Moreover, the curvature of fullerene molecules can be adjusted by the number of carbons and molecular symmetry[26]. Of these, $C_{60}$, with icosahedral ($I_h$) symmetry, not only possesses a curved surface but also exhibits unique electron-withdrawing properties[27]. It has recently been utilized as an electron buffer to balance the electronic density of active species[28] and as a building block for fullerene-LD nanohybrids to promote intermolecular electron transfer[29,30]. Besides, theoretical studies have demonstrated that charge transfer from metal to $C_{60}$ cage can enhance the adsorption of H on endohedral metallofullerenes ($M@C_{60}$) during hydrogen evolution[31]. In addition, the $sp^2$-hybridized carbon surface of $C_{60}$ is sufficiently reactive with metals due to the presence of electron-deficient olefinic $C = C$ bonds (i.e., more π-electron delocalization), to which metal atoms can bind in an $\eta^2$-$C_{60}$ π-type bonding mode[32–34]. $C_{60}$ has 30 $C = C$ bonds on its curved surface, which in theory could bind multiple metal atoms. The direct bonding of $C_{60}$ molecules to metal single atoms could immobilize the metal single atoms, avoiding the formation of aggregated particles. Moreover, the strong covalent interactions could accelerate the electron transfer process between metal and $C_{60}$, thereby optimizing the energy adsorption barriers for water and intermediates. Therefore, $C_{60}$ is an ideal support structure to anchor metal single atoms. Metal-$C_{60}$ polymer nanocatalysts have been successfully synthesized and showed high catalytic performance

for the reduction of quinoline and nitrobenzene[35–37]. However, the application of $C_{60}$ engineered single-atomic metal catalysts in the field of electrocatalysis and their experimental catalytic performances have never been explored.

In this work, we use $C_{60}$ as an electron-accepting support to anchor Pt atoms and synthesize highly-loaded and highly-dispersed Pt catalysts (Pt/$C_{60}$) at room temperature, whose inner structure is a Pt-$C_{60}$ polymer and the Pt sites are confined by two $C_{60}$ molecules. The resulting Pt/$C_{60}$-2 catalyst has a Pt loading as high as 21 wt%, with a majority of Pt single atoms and a minority of Pt clusters, which contribute to an extraordinary HER catalytic activity with a remarkably low onset potential and low overpotential of 25 mV at 10 mA cm$^{-2}$ in 1 M KOH solution. Density functional theory (DFT) calculations show that the Pt/$C_{60}$-2 catalyst promotes the adsorption of water, and the spherical shell-like electron cloud around Pt induced by the curved surfaces of two adjacent $C_{60}$ molecules in a $C_{60}$-Pt-$C_{60}$ configuration forms a built-in electric field that radiates from the central Pt atom to facilitate the desorption of hydrogen on Pt sites, thereby strongly promoting the electrocatalytic hydrogen evolution.

## Results
### Synthesis of Pt/$C_{60}$ catalysts
The preparation of Pt/$C_{60}$ followed previously described procedures[38,39]. As illustrated in Fig. 1, the toluene solution of Pt(dba)$_2$ (bis(dibenzylideneacetone) platinum) was slowly dropped into the $C_{60}$ toluene solution (molar ratio: n ($C_{60}$): n (Pt(dba)$_2$) = 1:1, 2:1, and 4:1, respectively) with vigorous stirring to inhibit the aggregation of Pt atoms. The reaction proceeded at room temperature under nitrogen for 60 h, yielding black precipitates. The black precipitates were collected by filtration and washed with toluene to remove residual Pt(dba)$_2$ and $C_{60}$. The precipitates were then dried under vacuum at 60 °C overnight to obtain the Pt/$C_{60}$ samples. Depending on the proportion of $C_{60}$ during the sample synthesis, the synthesized catalysts are defined as Pt/$C_{60}$–1, Pt/$C_{60}$-2, and Pt/$C_{60}$–4, respectively. The Pt/$C_{60}$ products are oligomeric in nature based on the facts that (i) the FT-IR absorption bands of Pt/$C_{60}$-2 contain only the characteristic peaks of $C_{60}$ but not those of dba, and (ii) the $C = O$ bond of Pt(dba)$_2$ is absent in the XPS C1s spectrum of Pt/$C_{60}$-2 (see Structure Characterizations section below), indicating that the ligands of Pt(dba)$_2$ were all substituted by $C_{60}$ during the reaction. Pt/$C_{60}$ catalysts thus have platinum-$C_{60}$ polymeric structures. And the Pt atoms are mainly in a divalent form (see Raman, XPS, and XAS results below), indicating the atomically dispersed Pt single atoms predominate and prefer to be

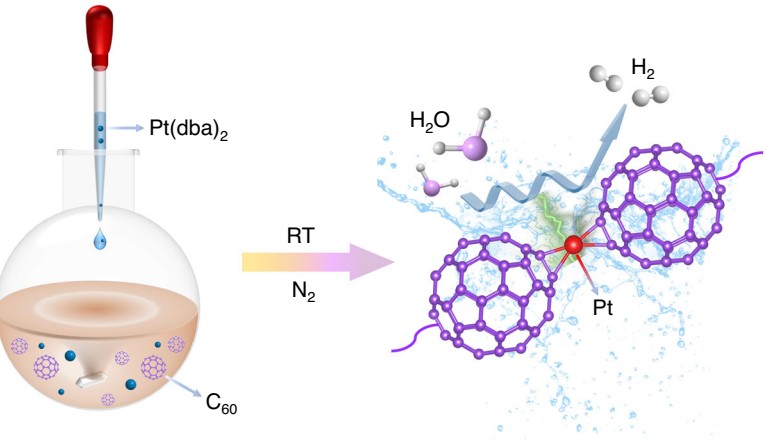

**Fig. 1 | Synthetic scheme of Pt/$C_{60}$ catalysts.** The toluene solution of Pt(dba)$_2$ and $C_{60}$ (molar ratio: n ($C_{60}$): n (Pt(dba)$_2$) = 1:1, 2:1, and 4:1, respectively) was stirred at room temperature under nitrogen, in which the dba ligands were completely substituted by $C_{60}$, forming a platinum-$C_{60}$ polymeric structure. The increased input of $C_{60}$ is to better disperse the Pt during the synthesis process and prevent the aggregation of Pt, and any excess $C_{60}$ was washed off after the reaction.

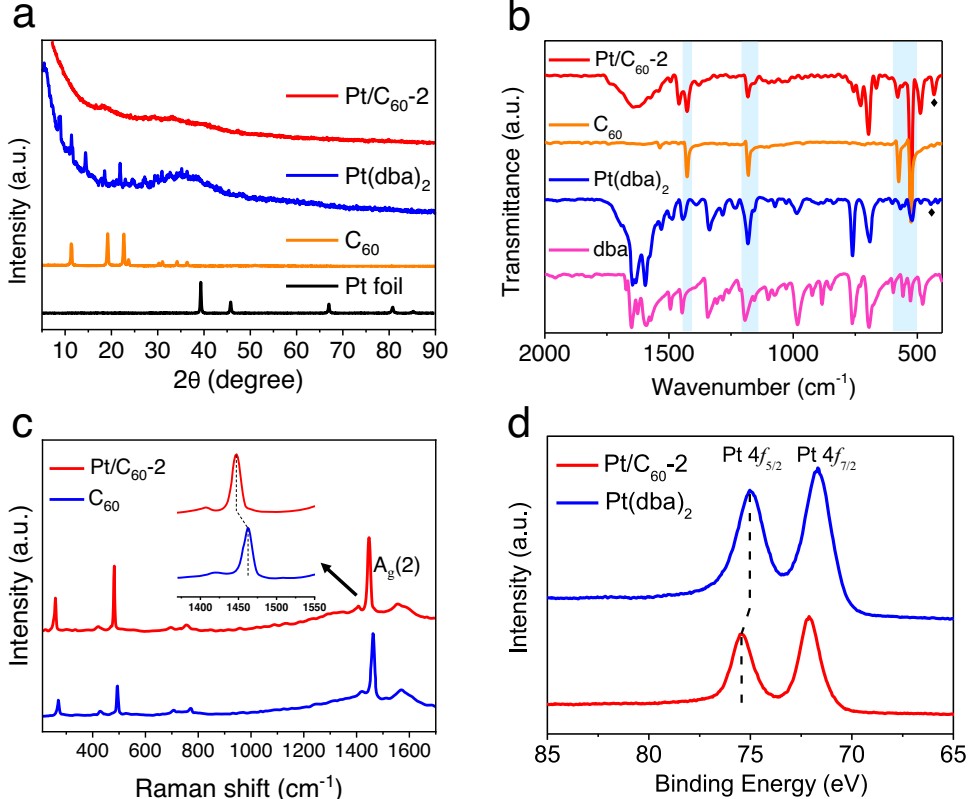

**Fig. 2 | Spectra analysis. a** PXRD patterns of Pt/C$_{60}$-2, Pt(dba)$_2$, C$_{60}$ and Pt foil. **b** IR spectra of Pt/C$_{60}$-2, C$_{60}$, Pt(dba)$_2$, and dba. **c** Raman spectra of Pt/C$_{60}$-2 and C$_{60}$. **d** High-resolution XPS Pt 4$f$ spectra of Pt/C$_{60}$-2 and Pt(dba)$_2$.

tetracoordinated[40,41]. Therefore, one Pt binds to two C$_{60}$ in an $\eta^2$-C$_{60}$ π-type bonding mode (Fig. 1). Based on inductively coupled plasma-optical emission spectrometry (ICP-OES) analysis, the content of Pt is 26.56 wt%, 21.54 wt%, and 20.48 wt% for Pt/C$_{60}$−1, Pt/C$_{60}$-2, and Pt/C$_{60}$-4, respectively (Supplementary Table 1).

**Structure characterizations of Pt/C$_{60}$-2**

In order to ascertain the structure of Pt/C$_{60}$-2 and the formation of a Pt-C$_{fullerene}$ bond in Pt/C$_{60}$-2, Powder X-ray diffraction (PXRD) measurements were performed. As shown in Fig. 2a, the characteristic peaks of Pt(dba)$_2$ and C$_{60}$ were not detected in Pt/C$_{60}$-2, indicative of the presence of new products. Moreover, the absence of the characteristic peaks of C$_{60}$ and metallic Pt indicated that the Pt/C$_{60}$ presents amorphous and there are not many aggregated Pt species in Pt/C$_{60}$-2, suggesting that high-loading atomic Pt on C$_{60}$ surface could be obtained, although we cannot exclude some ultrasmall Pt clusters from PXRD results. The absorption bands of Pt/C$_{60}$-2 in the FT-IR spectrum (Fig. 2b) (486, 525, 578, 666, 697, 726, 736, 755, 1183, 1425, and 1460 cm$^{-1}$) include the characteristic peaks of C$_{60}$ (525, 578, 1183 and 1435 cm$^{-1}$, highlighted by blue rectangular areas) but not those of Pt(dba)$_2$ or dba, indicating the presence of the intact C$_{60}$ structure and the absence of Pt(dba)$_2$ or dba moiety in Pt/C$_{60}$-2, thus ruling out the formation of C$_{60}$Pt(dba)$_2$ monomer. Notably, compared with the spectrum of dba, the FT-IR spectrum of Pt(dba)$_2$ (Fig. 2b) shows a new weak vibration at 443 cm$^{-1}$ (labeled with a rhombus), which can be assigned to the Pt-C coordination bond. Similarly, the FT-IR spectrum of Pt/C$_{60}$-2 also shows a vibration at 437 cm$^{-1}$ (labeled with a rhombus), but it is red-shifted and more intense compared to the corresponding vibration for Pt(dba)$_2$, suggesting that C$_{60}$ and Pt are covalently bonded in Pt/C$_{60}$-2. The conversion of Pt-C coordination bonds to Pt-C$_{fullerene}$ covalent bonds resulted in a reasonable shift of the corresponding IR absorption band due to the conjugation effect. Consistent

results were obtained from the Raman spectra (Fig. 2c), where Pt/C$_{60}$-2 displays identical peaks as C$_{60}$, except for the downshift of the A$_g$(2) peak, indicating the presence of the C$_{60}$ structure in Pt/C$_{60}$-2 and the interaction between Pt and C$_{60}$. As well known, the A$_g$(2) mode of C$_{60}$ in the Raman spectrum is sensitive to chemical modifications and is commonly accepted as an indicator for charge transfer between metal and C$_{60}$. As one electron is transferred to C$_{60}$, the A$_g$(2) mode is downshifted by ca. 6 cm$^{-1}$[37,42,43]. The A$_g$(2) peak of Pt/C$_{60}$-2 (1448 cm$^{-1}$) decreased by 14 cm$^{-1}$ relative to that of pure C$_{60}$ (1462 cm$^{-1}$) (Fig. 2c), indicating ~2.3 electrons were transferred from Pt to C$_{60}$ to form Pt-fulleride, which has a structure with each Pt atom connected to two neighboring C$_{60}$ molecules via a $\eta^2$-C$_{60}$ π-type bonding. The formation of Pt-C$_{fullerene}$ bonds in Pt/C$_{60}$-2 and the valence state of the Pt single atoms were further confirmed by X-ray photoelectron spectroscopy (XPS) analyses (Fig. 2d). The Pt 4$f$ XPS spectrum of Pt/C$_{60}$-2 was deconvoluted into two peaks, which are positively shifted relative to those for Pt(dba)$_2$, indicating that the oxidation state of Pt atoms increases upon the formation of Pt-C$_{fullerene}$ bonds. Moreover, the binding energies of the two peaks are 75.43 and 72.05 eV, respectively, corresponding to the 4$f_{5/2}$ and 4$f_{7/2}$ orbitals of Pt$^{2+}$ species[44], suggesting that Pt atoms bind to two C$_{60}$ in an $\eta^2$-C$_{60}$ π-type bonding mode. Calculated from the XPS results, the Pt loading reaches up to 21 wt%, which is consistent with the ICP-OES results (21.54 wt%, Supplementary Table 1) and beyond the benchmark Pt/C. In addition, the C = O bond of Pt(dba)$_2$ with an energy of 287.5 eV is absent in the C 1s spectrum of Pt/C$_{60}$-2 (Supplementary Fig. 1), providing additional evidence for the Pt/C$_{60}$-2 conversion from Pt(dba)$_2$ and C$_{60}$.

The morphology of Pt/C$_{60}$-2 was examined by scanning electron microscopy (SEM), transmission electron microscopy (TEM), and high-angle annular dark-field scanning transmission electron microscopy (HAADF-STEM). SEM image shows a spherical-like appearance of Pt/C$_{60}$-2 (Fig. 3a) and exhibits amorphous features as displayed in the

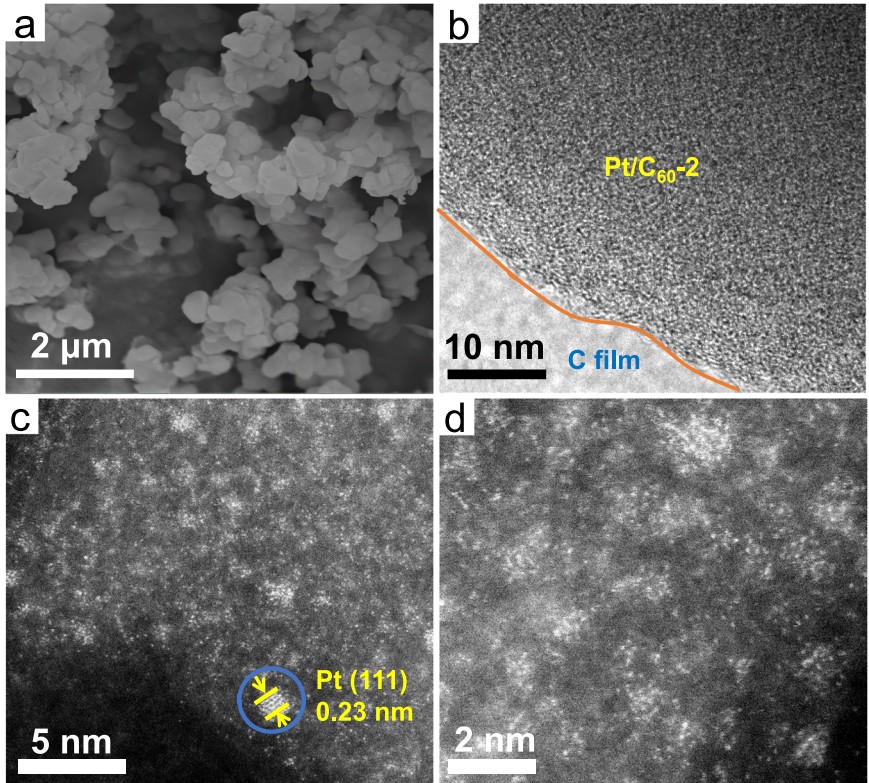

**Fig. 3 | Morphological characteristics of Pt/C$_{60}$-2. a** SEM, **b** HRTEM, and **c**, **d** HAADF-STEM images of Pt/C$_{60}$-2.

TEM image (Fig. 3b and Supplementary Fig. 2). The HAADF-STEM image and the corresponding EDX mappings confirmed that Pt/C$_{60}$-2 is composed of Pt and C elements (Supplementary Fig. 2d–f). Since the classical TEM analysis is challenging for single atoms and ultrasmall clusters, the specific form of the Pt component was further identified by atomic-resolution HAADF-STEM images. As expected, a large number of Pt single atoms were observed, together with some Pt clusters (Fig. 3c, d and Supplementary Figs. 3 and 4). The average size of Pt clusters is about 1.7 nm. Although both Pt single atoms and Pt clusters were detected in the sample, HAADF-STEM images taken in different regions indicated mainly the presence of atomically dispersed Pt (Supplementary Fig. 5). The predominance of Pt single atoms was further confirmed by cyclic voltammetry (CV) and X-ray absorption spectroscopy (XAS) measurements. CV curves of metal single-atom catalysts showed no features associated with hydrogen adsorption-desorption due to the oxidation state of metal species[45]. As displayed in Supplementary Fig. 6, unlike Pt/C, which is dominated by Pt nanoparticles (Supplementary Fig. 7), the CV curve of Pt/C$_{60}$-2 exhibits no hydrogen adsorption-desorption characteristic peaks, indicating that Pt atoms in Pt/C$_{60}$-2 are mainly atomically dispersed.

The oxidation state of the Pt atoms and the predominance of atomically dispersed Pt single atoms in Pt/C$_{60}$-2 were further verified by X-ray absorption spectroscopy (XAS) measurements (Fig. 4). The X-ray absorption near-edge structure (XANES) spectra of Pt L$_3$-edge is shown in Fig. 4a. The white line (WL) is related to the unoccupied density of states of the Pt 5$d$ orbitals[46]. The strong WL of Pt/C$_{60}$-2 suggests increased $d$-orbital vacancies for Pt as well as an elevated oxidation state for Pt compared to Pt foil, which agrees well with the Raman and XPS results. The coordination environment of Pt was examined by extended X-ray absorption fine structure (EXAFS) analyses. In clear contrast, Pt/C$_{60}$-2 shows a dominant peak at 1.68 Å, which is consistent with the Pt-C coordination[47], confirming the bond (Pt-C$_{fullerene}$) formation between Pt and C$_{60}$, while the small peak at 2.56 Å may arise from Pt-Pt bonds (Fig. 4b). Compared with the peak

intensity corresponding to Pt-C coordination, the significantly weaker peak of Pt-Pt coordination indicates a smaller proportion of Pt clusters and the atomically dispersed nature of Pt species in Pt/C$_{60}$-2, consistent with the TEM and CV results. The coordination environment of Pt was further verified by EXAFS wavelet transform (EXAFS-WT) analysis (Fig. 4d-f). The intense peaks of Pt foil, Pt/C$_{60}$-2, and PtO$_2$ are located at 8.4, 3.9, and 4.6 Å, corresponding to Pt-Pt, Pt-C, and Pt-O bonds, respectively[48], showing that Pt atoms and C$_{60}$ molecules are connected by Pt-C bonds. The Pt-Pt bond was not observed by EXAFS-WT for Pt/C$_{60}$-2, indicative of insignificant amounts of Pt clusters in the sample. From the fitting curve (Fig. 4c), the coordination number of Pt was estimated to be ca. 3 (Supplementary Table 2). Since there are 4 Pt-C bonds for Pt single atom sites, a coordination number of 3 indicates that Pt clusters also connect to C$_{60}$ to lower the coordination number. We can thus speculate that atomically isolated Pt is coordinated by four carbon atoms from two C$_{60}$ molecules to form a Pt−C$_4$ configuration, while the Pt atoms connected to C$_{60}$ in the Pt clusters are coordinated only to two carbon atoms (inset of Fig. 4c). A coordination number of 3 indicates that approximately 75% of Pt is bonded to C in the Pt/C$_{60}$-2 sample, while ca. 25% is the metallic Pt$^0$. Combined with the HAADF-STEM results, it can be concluded that atomically dispersed Pt single atoms are dominant in Pt/C$_{60}$-2. Therefore, we can speculate that Pt/C$_{60}$-2 is a high-loading Pt catalyst whose inner structure can be described as a platinum-C$_{60}$ polymeric structure, in which the predominant Pt single atoms serve as bridges connecting the adjacent C$_{60}$, and Pt clusters are anchored on C$_{60}$ as well. These combined results demonstrate a very high-loading and highly dispersed Pt catalyst resulting from a room-temperature chemical reaction between C$_{60}$ and the Pt precursor.

### Electrocatalytic HER performance of Pt/C$_{60}$-2

To the best of our knowledge, most of the Pt single atoms present in the planar configurations of Pt-N$_4$[6,48], Pt-C$_3$[49], Pt-C$_2$N$_1$[44], and Pt-N$_1$C$_3$[48], Pt-C$_4$ located on two C$_{60}$ curved surfaces have never been studied for

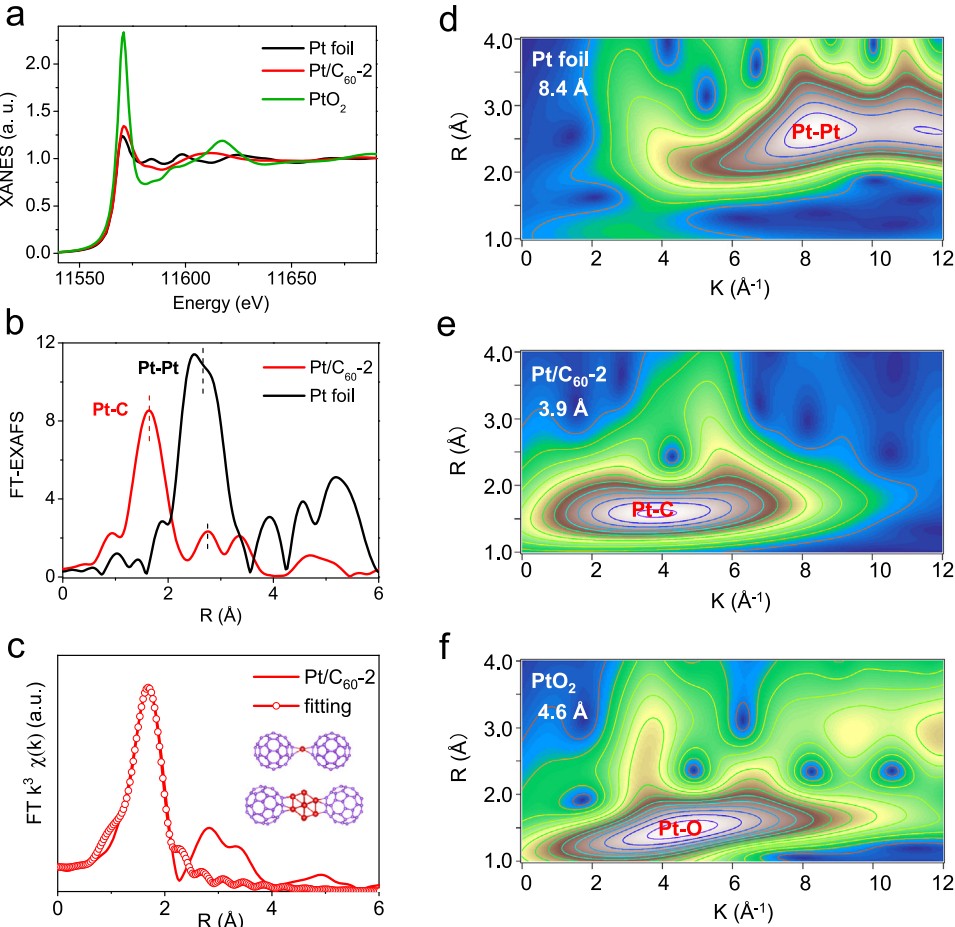

**Fig. 4 | X-ray absorption spectroscopic studies. a** XANES spectra of the Pt $L_3$-edge of Pt/$C_{60}$-2, Pt foil and PtO$_2$ **b** FT-EXAFS spectra of the Pt $L_3$-edge of Pt/$C_{60}$-2 and Pt foil. **c** Corresponding EXAFS fitting curve for the Pt/$C_{60}$-2. The inset shows the structural models for Pt/$C_{60}$-2. **d−f** Wavelet transforms for the $K^3$-weighted EXAFS of Pt/$C_{60}$-2 and the reference samples for Pt foil and PtO$_2$.

alkaline HER. To investigate the HER activity of Pt/$C_{60}$-2, linear sweep voltammetry (LSV) curves for Pt/$C_{60}$-2/KB (Ketjen Black to improve conductivity), KB, $C_{60}$ physically mixed with Pt(dba)$_2$ and KB, and for commercial 20 wt% Pt/C were measured in 1 M KOH at a scan rate of 5 mV$^{-1}$ with 90% iR compensation. (The LSV curves without iR compensation are shown in Supplementary Fig. 8a). As shown in Fig. 5a, the Pt/$C_{60}$-2/KB displays a remarkably low overpotential of 25 mV to reach the current density of 10 mA cm$^{-2}$, lower than the value of 39 mV for 20 wt% Pt/C (Supplementary Fig. 8b) and outperforms most state-of-the-art alkaline Pt-based catalysts thus far reported (Supplementary Table 3). Moreover, Pt/$C_{60}$-2/KB shows a higher mass activity than Pt/C (Supplementary Fig. 8c). In contrast, the catalytic activity of physically mixed $C_{60}$/Pt(dba)$_2$/KB decreased considerably when compared to that of Pt/$C_{60}$-2, demonstrating the crucial role of $C_{60}$ and Pt bonding. The coordination of Pt atoms by $C_{60}$ controls the electronic structure of Pt and results in excellent catalytic activity, which was confirmed by theoretical calculations, as described in the DFT section below. The HER mechanism on the Pt/$C_{60}$-2/KB electrocatalyst was probed by Tafel slopes (Fig. 5b). The slope value of Pt/$C_{60}$-2/KB is 55 mV dec$^{-1}$, much smaller than the 99 mV dec$^{-1}$ for Pt/C, confirming a typical Volmer-Heyrovsky mechanism and faster HER reaction kinetics. By extrapolating the Tafel plots to the point where the overpotential is equal to 0 V, the exchange current density (j$_0$) of Pt/$C_{60}$-2/KB was calculated to be 2.8 mA cm$^{-2}$, higher than that of Pt/C (2.5 mA cm$^{-2}$) (Supplementary Fig. 8b), confirming the high activity of Pt/$C_{60}$-2/KB.

The charge transfer properties of Pt/$C_{60}$-2/KB were analyzed using electrochemical impedance spectroscopy (EIS), and compared with those of $C_{60}$/Pt(dba)$_2$/KB and 20 wt% Pt/C (Fig. 5c). As observed from the Nyquist plots, the Nyquist semicircle diameters, representing the charge transfer resistance ($R_{ct}$), show a decreasing trend from $C_{60}$/Pt(dba)$_2$/KB (67.8 Ω) to 20 wt% Pt/C (51.5 Ω) to Pt/$C_{60}$-2/KB (45 Ω), consistent with the LSV results (Fig. 5a). The EIS results evidence the strong interactions between the Pt atoms and the $C_{60}$ molecules, which lead to a rapid electron transfer rate at the interface between the catalyst and the electrolyte. As reflected by electrochemical double layer capacitances ($C_{dl}$) (Fig. 5d) derived from the cyclic voltammetry (CV) curves (Supplementary Fig. 9), Pt/$C_{60}$-2/KB possesses the highest $C_{dl}$ value of 22 mF cm$^{-2}$ among all the samples, signaling the largest electrochemically active surface area (ECSA) and more accessible active sites in Pt/$C_{60}$-2/KB. To evaluate the intrinsic activity of Pt/$C_{60}$-2, turnover frequency (TOF), defined as the number of molecules (H$_2$) produced per site per second, for Pt/$C_{60}$-2/KB and Pt/C were calculated through the characterization of catalyst surface area via CO-stripping method[50]. As depicted in Fig. 5e, the TOF values for Pt/$C_{60}$-2/KB are higher than those for Pt/C in a wide range of overpotential. For example, at 50 mV, 100 mV, and 150 mV, the TOF values are 2.17 s$^{-1}$, 5.55 s$^{-1}$, and 11.2 s$^{-1}$, respectively for Pt/$C_{60}$-2, which are almost twice the Pt/C values, and are comparable or even superior to the values of the reported catalysts (Fig. 5e and Supplementary Table 4), indicating a high hydrogen production efficiency for Pt/$C_{60}$-2/KB.

Durability is also an important parameter for high-efficiency electrocatalysts[51]. The cyclic stability of Pt/$C_{60}$-2/KB and Pt/C were first evaluated by comparing the LSV curves. Pt/$C_{60}$-2/KB shows negligible decay in catalytic activity after 3000 cyclic voltametric

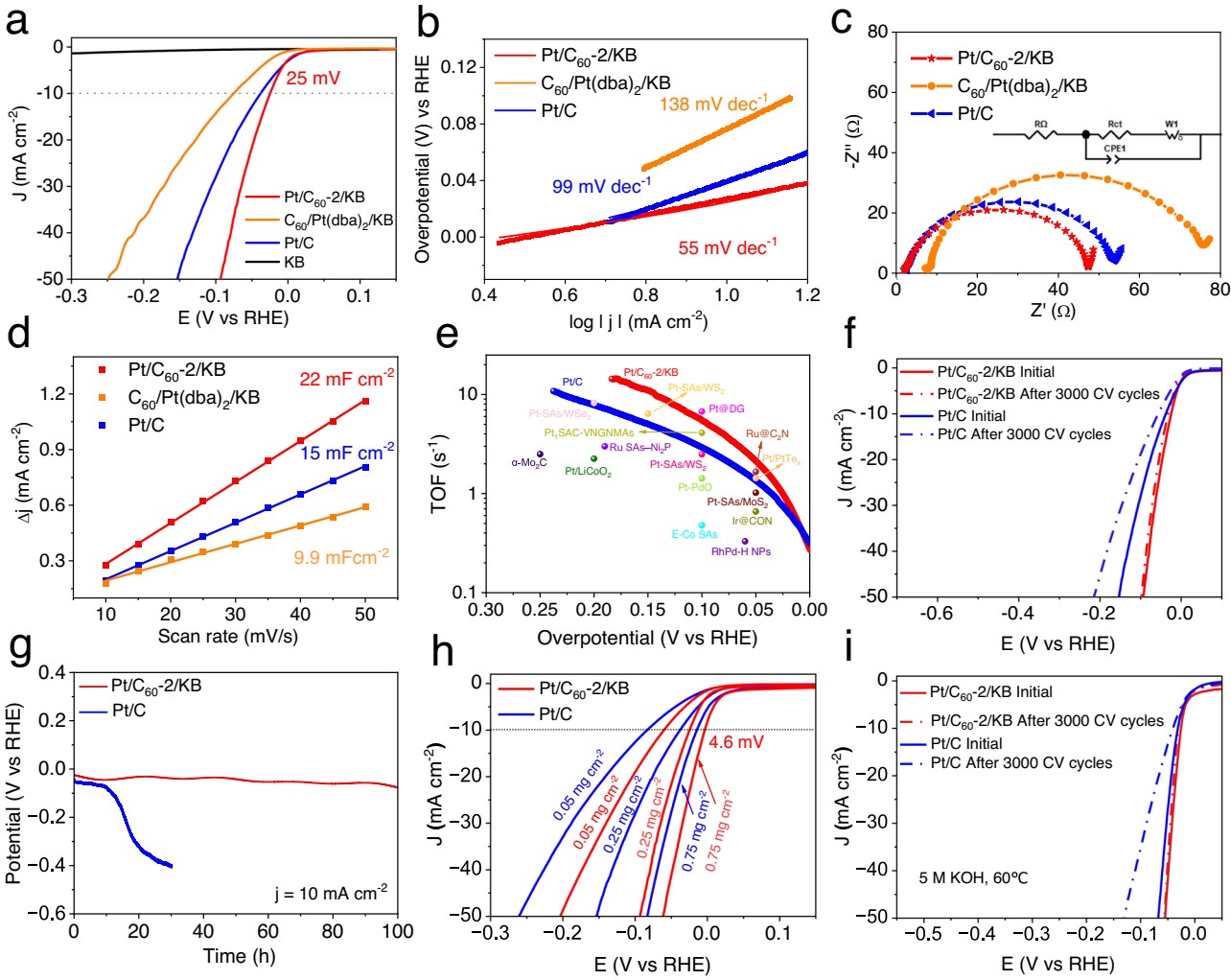

**Fig. 5 | HER performance of Pt/C$_{60}$-2 and commercial Pt/C. a** HER polarization curves for Pt/C$_{60}$-2/KB, C$_{60}$/Pt(dba)$_2$/KB, Pt/C, and KB in 1 M KOH. **b** Tafel plots derived from the corresponding polarization curves. **c** Nyquist plots for Pt/C$_{60}$-2/KB, C$_{60}$/Pt(dba)$_2$/KB, and Pt/C at an overpotential of 1.5 V (vs RHE) (The system resistance is 2.5 Ω and 2.6 Ω for Pt/C$_{60}$-2/KB and Pt/C measurements, respectively). **d** Capacitive Δ j = j$_a$ − j$_c$ as a function of scan rate in 1 M KOH. **e** TOF values for Pt/C$_{60}$-2/KB, Pt/C and other reported electrocatalysts over a wide range of overpotentials. **f** LSV curves for Pt/C$_{60}$-2/KB and Pt/C before and after 3000 catalytic cycles in 1 M KOH. **g** Long-term stability test at a current density of 10 mA cm$^{-2}$ in 1 M KOH. **h** HER polarization curves for Pt/C$_{60}$-2/KB and Pt/C under multiple loadings in 1 M KOH. **i** LSV curves for Pt/C$_{60}$-2/KB and Pt/C before and after 3000 catalytic cycles in 5 M KOH at 60 °C (LSV curves are with 90% iR correction. Non-iR corrected data for Fig. 5a, h, i is provided as Supplementary Fig. 11).

sweeps, whereas obvious attenuation is observed for 20 wt% commercial Pt/C (Fig. 5f). The long-term stability for 100 h at 10 mA cm$^{-2}$ in 1 M KOH was monitored and the overpotential only increased 31.5 mV, much better than that for Pt/C (Fig. 5g). TEM images showed that the spherical morphology of Pt/C$_{60}$-2 was retained after the stability test (Supplementary Fig. 10a–d). Moreover, Pt single atoms are clearly observed in the HAADF-STEM images (Supplementary Fig. 10e), indicating that Pt single atoms are very stable during catalysis. Consistent results were obtained from XPS since the peak positions did not change relative to those before the catalysis (Supplementary Fig. 10f). Although more Pt clusters were formed during the long-term durability test, the dimensions of the clusters remained unchanged (Supplementary Fig. 10d, e). To determine Pt leaching after the stability test, inductively coupled plasma-mass spectrometry (ICP-MS) was conducted to analyze the electrolyte and the concentration of Pt ions was determined to be only 0.034 mg L$^{-1}$, further indicating the robust nature of Pt/C$_{60}$-2.

In addition, the activities of Pt/C$_{60}$-2/KB and Pt/C were further assessed under multiple loadings. As shown in Fig. 5h, Pt/C$_{60}$-2 exhibits lower overpotentials than Pt/C under different loadings (Supplementary Fig. 8d). Moreover, by increasing the catalyst loading, the overpotentials are decreased, which is due to the increase in the number of active sites[50]. To further estimate the practicability of the catalyst, the catalytic performance of Pt/C$_{60}$-2/KB and Pt/C were examined in 5 M KOH at 60 °C. As displayed in Fig. 5i, Pt/C$_{60}$-2 shows a lower overpotential (η$_{10}$ = 26.1 mV) than Pt/C (η$_{10}$ = 32.1 mV) at the current density of 10 mA cm$^{-2}$. Moreover, after 3000 cyclic CV tests, the overpotential of Pt/C$_{60}$-2/KB at 20 mA cm$^{-2}$ increased only 4 mV, whereas a 23 mV increase is observed for Pt/C, indicating that Pt/C$_{60}$-2 still exhibits superior activity and stability over Pt/C under harsh conditions. Finally, the Faradaic efficiency of Pt/C$_{60}$-2/KB for HER was determined by quantifying the volume of the generated H$_2$ gas, which showed a > 98 % Faraday efficiency for 4 h (Supplementary Fig. 12).

## Insights into the nature of the active sites

As both Pt single atoms and Pt clusters are present in the Pt/C$_{60}$-2 samples, potassium thiocyanate (KSCN) and ethylenediaminetetraacetic acid (EDTA) were used as poisoning reagents to determine

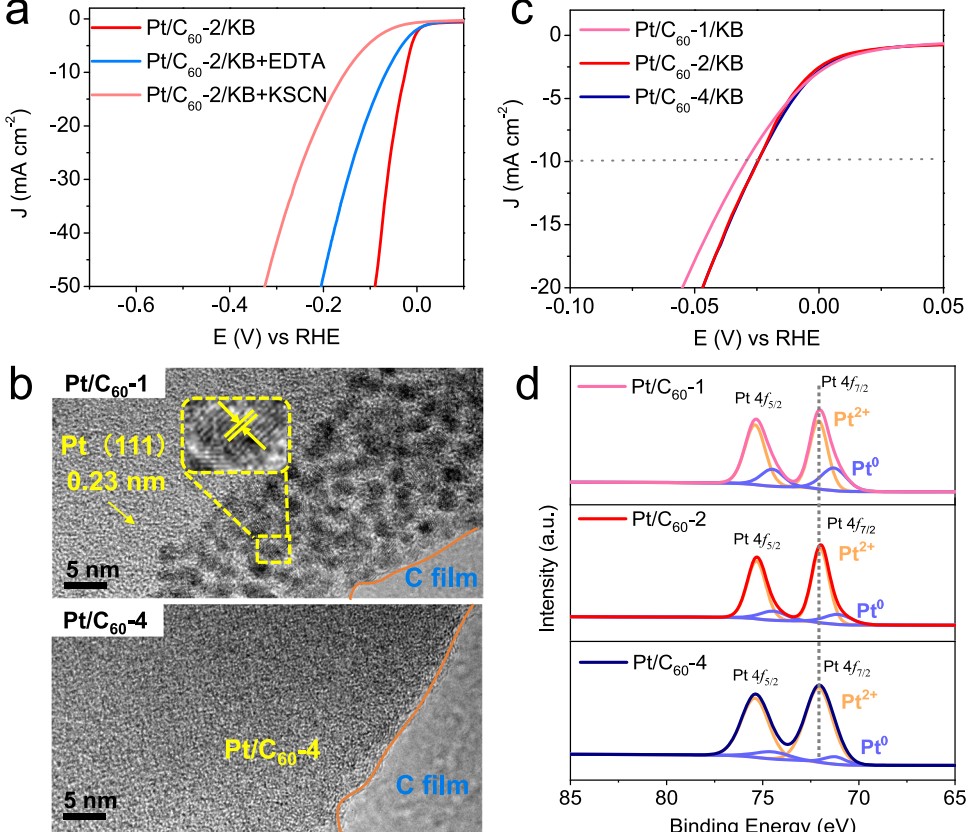

**Fig. 6 | Control experiments for studying the active sites. a** Catalyst poisoning experiments for Pt/$C_{60}$-2/KB in 1 M KOH with the addition of 10 mM EDTA or KSCN. **b** The HR-TEM images for the control samples Pt/$C_{60}$-1 and Pt/$C_{60}$-4. **c** HER polarization curves for control samples Pt/$C_{60}$-1 and Pt/$C_{60}$-4. **d** High-resolution XPS Pt 4$f$ spectra for Pt/$C_{60}$-1, Pt/$C_{60}$-2, and Pt/$C_{60}$-4.

their respective contributions[52]. SCN⁻ can inactivate both Pt clusters and Pt single atoms by adsorption onto the surface of Pt, while EDTA mainly deactivates Pt single atoms by coordinating with it. As the results show in Fig. 6a, the $\eta_{10}$ value of Pt/$C_{60}$-2/KB was negatively shifted by >120 mV with the addition of 10 mM KSCN into the KOH electrolyte, while the shift was only ca. 40 mV when 10 mM EDTA was added. The poisoning results suggested that, although both Pt clusters and Pt single atoms contributed to the HER activity, the Pt single atoms played a dominant role.

The predominant contribution of Pt single atoms toward the HER was further demonstrated by control experiments, in which the molar ratio of $C_{60}$ and Pt(dba)$_2$ during sample syntheses was controlled to be 1:1 and 4:1, and the corresponding samples were designated as Pt/$C_{60}$-1 and Pt/$C_{60}$-4, respectively. As the HR-TEM images show in Fig. 6b and Supplementary Fig. 13, both Pt/$C_{60}$-1 and Pt/$C_{60}$-4 show amorphous features, and compared to Pt/$C_{60}$-2 and Pt/$C_{60}$-4, the number of Pt clusters in Pt/$C_{60}$-1 is significantly increased due to the higher molar ratio of Pt(dba)$_2$:$C_{60}$ during sample synthesis. Consistent results were obtained from Pt 4$f$ XPS measurements, where the Pt⁰ fraction was gradually increased from 8.9 % for Pt/$C_{60}$-4 to 12.2 % for Pt/$C_{60}$-2, and 28.1 % for Pt/$C_{60}$-1 (Fig. 6d), indicating that the addition of more $C_{60}$ is helpful to better disperse Pt and decrease the Pt aggregation, although the excessive $C_{60}$ is washed away by toluene after the synthesis. Based on the ICP-OES results, the Pt loading for Pt/$C_{60}$-4 is ca. 20.48 wt%, which is almost the same as that for Pt/$C_{60}$-2 (21.54 wt%), thus delivering similar overpotentials. By contrast, although the Pt content for Pt/$C_{60}$-1 increases to 26.56 wt%, the number of Pt clusters increased as well, resulting in an increase of 6 mV (at 10 mA cm⁻²) in overpotential compared with those for Pt/$C_{60}$-2 and Pt/$C_{60}$-4 (Fig. 6c). These results further confirm that

atomically dispersed Pt is the dominant catalytically active sites with less catalytic contribution from the Pt clusters.

## Density functional theory calculations

To further understand the catalytic performance of synthesized Pt/$C_{60}$-2 and probe the origin of the remarkable HER catalytic activity, density functional theory (DFT) calculations were carried out. For a single Pt atom interacting with two $C_{60}$, it is preferably at the bridge position between two hexagonal rings ($\eta^2$), forming 4 Pt-C bonds. The $\eta^2$ coordination mode has been experimentally observed for $C_{60}$-Pt complexes[40,41]. For a metal cluster interacting with two $C_{60}$, there are three types of bonding modes[53], of which the bonding across the hexagonal face of $C_{60}$ ($\mu_3$-$\eta^2$: $\eta^2$: $\eta^2$-$C_{60}$) is the most stable mode[35,36]. And this type of bonding mode has been recognized in $C_{60}$-metal cluster complexes[54]. In view of the polymeric structure of Pt/$C_{60}$-2, the minimum periodic models for Pt single atoms ($C_{60}$-Pt-$C_{60}$) and Pt clusters ($C_{60}$-Pt$_{13}$-$C_{60}$) were established, respectively. In $C_{60}$-Pt-$C_{60}$, the Pt atom (marked as active site $C_{60}$-Pt-$C_{60}$-1) is bonded to four carbon atoms from two $C_{60}$ molecules (Fig. 7a). $C_{60}$-Pt$_{13}$-$C_{60}$ was built according to the Pt (111) crystal planes exposed by Pt clusters, in which there are two types of Pt atoms, one is connected to the $C_{60}$ cage (denoted as active site $C_{60}$-Pt$_{13}$-$C_{60}$-2), and the other is connected to adjacent Pt atoms (denoted as active site $C_{60}$-Pt$_{13}$-$C_{60}$-3) (Fig. 7b). The interactions between Pt atoms and $C_{60}$ cages in $C_{60}$-Pt-$C_{60}$ and $C_{60}$-Pt$_{13}$-$C_{60}$ were probed by charge density difference calculations. As shown in Fig. 7c-d, the charge density decreases on the Pt site, while it increases on the fullerene C atoms attached to Pt, indicative of charge transfer processes from Pt to the electron-withdrawing $C_{60}$, which is consistent with the XPS, Raman and EXAFS results. Especially for $C_{60}$-Pt-$C_{60}$, the electron cloud forms a spherical shell around the central Pt

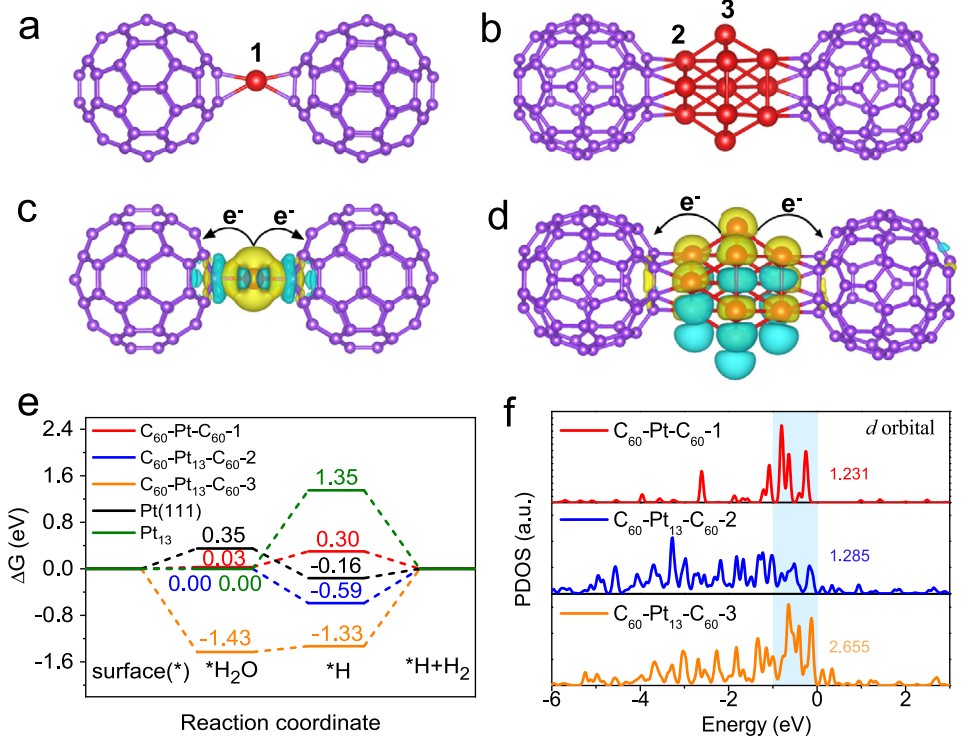

**Fig. 7 | Theoretical calculations of electronic structure and energy diagram for different Pt sites.** The side view of optimized model for **a** Pt single atom and **b** small Pt clusters. The charge differential density map of **c** $C_{60}$-Pt-$C_{60}$ and **d** $C_{60}$-$Pt_{13}$-$C_{60}$, blue and yellow represent electron density decrease and increase, respectively. **e** Calculated adsorption energies of $H_2O$ and H on the surface for $C_{60}$-Pt-$C_{60}$-1, $C_{60}$-$Pt_{13}$-$C_{60}$-2, $C_{60}$-$Pt_{13}$-$C_{60}$-3, Pt (111), and $Pt_{13}$ cluster. **f** PDOS for Pt atoms for $C_{60}$-Pt-$C_{60}$-1, $C_{60}$-$Pt_{13}$-$C_{60}$-2 and $C_{60}$-$Pt_{13}$-$C_{60}$-3.

atom due to the effects of two $C_{60}$ curved surfaces. Thus, a built-in electric field radiates from the central Pt atom to the surroundings, which is favorable for the desorption of hydrogen.

The alkaline HER mechanism is usually divided into three steps, namely $H_2O$ adsorption, $H_2O$ dissociation, and $H_2$ desorption[3]. Supplementary Fig. 14 displays the schematic diagram of the adsorption of *$H_2O$ and *H at three different active sites. The kinetic energy for the $H_2O$ adsorption ($\Delta G_{*H_2O}$), hydrogen adsorption ($\Delta G_{*H}$), and final $H_2$ release were calculated based on the optimized models for $C_{60}$-Pt-$C_{60}$, $C_{60}$-$Pt_{13}$-$C_{60}$, Pt (111), and $P_{13}$ cluster. The adsorption energies for $H_2O$ ($\Delta G_{*H_2O}$) and *H ($\Delta G_{*H}$) has been widely accepted as descriptors to evaluate the HER activity of catalysts. Catalysts with a smaller $\Delta G_{*H_2O}$ can speed up the reaction kinetics, and catalysts with $\Delta G_{*H}$ value close to 0 eV are considered to be ideal for HER[55]. As shown in Fig. 7e, the adsorption of $H_2O$ occurs almost spontaneously at three active sites for Pt/$C_{60}$-2 and $P_{13}$ cluster, while a high energy barrier of 0.35 eV needs to be overcome on the Pt (111) surface. Although $C_{60}$-$Pt_{13}$-$C_{60}$-3 shows the strongest *$H_2O$ adsorption with a $\Delta G_{*H_2O}$ value of −1.43 eV, the interaction between $C_{60}$-$Pt_{13}$-$C_{60}$-3 and *H is also rather strong ($\Delta G_{*H}$ = −1.33 eV), which is not conductive for efficient release of gaseous $H_2$. In contrast, $C_{60}$-Pt-$C_{60}$-1 displays moderate adsorption for both *$H_2O$ and *H, with the $\Delta G_{*H_2O}$ and $\Delta G_{*H}$ values of 0.03 eV and 0.30 eV, respectively. Although the adsorption of $H_2O$ on the $Pt_{13}$ cluster occurs spontaneously, the adsorption of *H on the $Pt_{13}$ cluster needs to overcome a high energy barrier of 1.35 eV, which greatly impedes the efficient formation of $H_2$ molecules. When $Pt_{13}$ was connected to $C_{60}$, the adsorption of *H was remarkably enhanced. Compared with $C_{60}$-$Pt_{13}$-$C_{60}$-2, $C_{60}$-$Pt_{13}$-$C_{60}$-3, and Pt (111) active sites, the atomic $C_{60}$-Pt-$C_{60}$-1 site exhibits the optimal HER kinetics. The results are in agreement with the Sabatier principle stating that interactions of

$H_2O$ and intermediate H with the catalytically active surfaces follow the principle of moderation rather than strong or weak[56].

In addition, the projected density of states (PDOS) for $d$-orbitals of Pt atoms were calculated to explain the different adsorption energies for water or hydrogen on the three active sites (Fig. 7f). By comparing the active electron densities of Pt-5$d$ near the Fermi level (highlighted by blue rectangular areas), a gradual increase in the numbers of electronic states from $d$ orbitals for $C_{60}$-Pt-$C_{60}$-1 to $C_{60}$-$Pt_{13}$-$C_{60}$-2 to $C_{60}$-$Pt_{13}$-$C_{60}$-3 was observed, corresponding to the binding intensity of $H_2O^*$ and $H^*$ from weak to strong[57], which are consistent with the results of free energy calculations. Therefore, the Pt single atom sites modified by $C_{60}$ enhance the water adsorption and hydrogen desorption, thus being the optimal active sites in Pt/$C_{60}$-2, consistent with the experimental results.

## Discussion

In summary, we have synthesized and characterized the structure and catalytic performance of highly-loaded, highly-dispersed $C_{60}$ confined Pt catalysts for accelerated HER. Pt/$C_{60}$-2 outperformed the benchmark Pt/C for alkaline HER in terms of activity and stability, which represents a significant advancement in developing high-performance catalysts by engineering single metal atoms on spherical fullerene supports. The excellent HER activity of Pt/$C_{60}$-2 was attributed to (1) the highly-loaded, highly-dispersed atomic Pt active sites, which maximize the utilization of Pt atoms; (2) the charge redistribution between Pt and $C_{60}$, which delivers a favorable adsorption energy towards *$H_2O$ and *H. This study not only fills a gap in the application of metal-fulleride polymers in the field of electrocatalysis but also provides a new perspective for designing efficient single-atom catalysts. The large portfolio of fullerene molecules and metal species that are available pave the way for a new generation of single-atom catalysts.

## Methods

### Synthesis of Pt(dba)$_2$

The precursor, bis(dibenzylideneacetone) platinum (Pt(dba)$_2$), was prepared according to a previous report[58]. Sodium acetate (200 mg, 2.44 mmol) and dibenzylideneacetone (dba) (160 mg, 2.44 mmol) were dissolved in ethanol (12.5 mL) by stirring in air. Finely ground K$_2$PtCl$_4$ (100 mg, 0.813 mmol) was dissolved in DI water (7.5 mL) and then slowly added to the above ethanol solution. The mixture was refluxed at 90 °C for 24 h under a nitrogen atmosphere, and a purple-black precipitate was formed. The precipitate was filtered and washed with ethanol and DI water three times to remove the sodium salts and unreacted dba. Afterward, the crude product was poured into 5 mL THF under sonication for 10 min to dissolve the dba trapped inside, and then 250 mL ethanol was slowly added to re-precipitate Pt(dba)$_2$. Finally, the product was isolated by filtration, washed with ethanol, and dried under vacuum at 60 °C overnight. The structure was confirmed by NMR spectroscopy (Supplementary Fig. 15), which is consistent with that reported in the literature[59].

### Synthesis of Pt/C$_{60}$-2

36 mg C$_{60}$ (50 mmol) was completely dissolved in 23 mL toluene (dried with molecular sieves) under sonication. Similarly, 16.6 mg Pt(dba)$_2$ (25 mmol) was dissolved in 30 mL toluene to form a clear solution (the molar ratio of C$_{60}$ to Pt(dba)$_2$ was 2:1). The toluene solution of Pt(dba)$_2$ was slowly dropped into the C$_{60}$ toluene solution under vigorous stirring. After the mixture was stirred at room temperature for 60 h under nitrogen, black precipitates were obtained and collected by filtration. Then the precipitates were washed with toluene until the filtrate was colorless and transparent. Finally, the precipitates were dried under vacuum at 60 °C overnight to obtain Pt/C$_{60}$-2. Samples of Pt/C$_{60}$-1 and Pt/C$_{60}$-4 were prepared by the same procedure except that the molar ratio of C$_{60}$ to Pt(dba)$_2$ was 1:1 and 4:1, respectively, during sample synthesis.

### Material characterizations

Scanning electron microscopy was performed on an FEI nova 450. Transmission electron microscopy and STEM-EDX were performed on a Tecnai G2 F30 field-emission microscope operated at an accelerating voltage of 200 kV. Spherical aberration-corrected STEM images were obtained using a 200 kV JEM-ARM200F equipped with double spherical aberration correctors, a 300 kV FEI-Titan ETEM G2 80-300 equipped with double spherical aberration correctors, and a 300 kV ThermoFisher Themis Z equipped with double spherical aberration correctors. XPS measurements were carried out on a Thermo Scientific K-Alpha spectrometer equipped with an Al anode (Al Kα = 1,486.7 eV). The Avatage software was used for the analysis of the XPS spectra. FT-IR spectra were obtained using a Fourier transform infrared spectrometer (vertex70). X-ray diffraction patterns were recorded by on a Rigaku Smartlab instruments equipped with Cu Kα radiation (λ = 1.54178 Å). Raman spectra (tested with LabRAM HR800) were obtained to determine the structures and compositions of the samples. ICP-OES was performed on Agilent 730, and ICP-MS was conducted on PerkinElmer NexION 350X.

### XAS measurements and simulations

XANES is the measurement of the X-ray absorption coefficient of a material as a function of energy, typically in a 50-100 eV range that begins before the absorption edge of a given element in the sample. The Pt L$_3$ edge XAFS spectra were collected at the beamline 14W1 in the Shanghai Synchrotron Radiation Facility, a 3.5 GeV third-generation synchrotron source, using a Si (111) double-crystal monochromator in transmission mode. The data processing was carried out through the Athena program implemented in Demeter software[60]. By deducting the post-edge background from the total absorption and normalizing

with respect to the edge-jump step, we obtained the $k^3$-weighted EXAFS data. In order to separate the EXAFS contributions from different coordination shells, $k^3$-weighted $\chi(k)$ data of Pt L$_3$-edge were Fourier transformed to real (R) space using a hanning windows ($d$k = 1.0 Å$^{-1}$). The data fitting was performed using the Artemis module of Demeter software packages.

### Electrochemical measurements

To fabricate the working electrode, 5 mg of the catalyst and 2.5 mg ketjen black with 60 μL of Nafion solution were dispersed in 1440 μL of a water/isopropanol mixed solvent (1:1 volume ratio) over 2 h of sonication to form a homogeneous ink. 15 μL catalyst ink was dropped onto the surface of a glassy carbon disk working electrode. The loading of Pt/C$_{60}$-2 was 0.25 mg/cm$^2$ (0.054 mg(Pt)/cm$^2$) for all electrochemical tests in 1 M KOH. For the Pt/C working electrode, 5 mg of the catalyst with 40 μL of Nafion solution were dispersed in 960 μL of a water/isopropanol mixed solvent (1:1 volume ratio) over 2 h of sonication to form a homogeneous ink. Then 10 μL catalyst ink was dropped onto the surface of a glassy carbon disk working electrode. The loading of Pt/C was 0.25 mg/cm$^2$ (0.050 mg(Pt)/cm$^2$) for all electrochemical tests in 1 M KOH. The glassy carbon disk electrode of the RDE was 5 mm in diameter. Notably, to evaluate the stability of the Pt/C$_{60}$-2 catalyst after ink preparation, structure characterizations of the ink sample were conducted. The PXRD, IR, Raman, and XPS results of the ink (Pt/C$_{60}$-2/KB) are consistent with those of pristine Pt/C$_{60}$-2 (Supplementary Fig. 16a–d). Moreover, the atomically dispersed Pt single atoms are observed in the ink sample (Supplementary Fig. 16e, f), confirming that the Pt single atoms remained after ink preparation, and the structure of Pt/C$_{60}$-2 is preserved. In addition, to assess the tolerance of Pt/C$_{60}$-2 against 1 M KOH, we leave Pt/C$_{60}$-2 in 1 M KOH overnight under stirring. The IR absorption bands of the KOH-treated Pt/C$_{60}$-2 are almost the same as that of fresh Pt/C$_{60}$-2, and the absorption band of O-H was not observed for Pt/C$_{60}$-2 (Supplementary Fig. 17), indicating the absence of fullerol and the excellent tolerance of Pt/C$_{60}$-2 to KOH.

For the electrochemical tests in 5 M KOH at 60°C, the working electrode was fabricated as follows: 5 mg of the catalyst Pt/C$_{60}$-2 and 2.5 mg ketjen black with 60 μL of Nafion solution were dispersed in 1440 μL of a water/isopropanol mixed solvent (1:1 volume ratio) over 1 h of sonication to form a homogeneous ink. Then 24 μL catalyst ink was dropped onto the surface of a carbon cloth (0.4 × 0.4 cm$^2$), thus the loading of Pt/C$_{60}$-2 was 0.5 mg/cm$^2$ (0.108 mg(Pt)/cm$^2$). For the Pt/C electrode, 5 mg of Pt/C with 40 μL Nafion solution were dispersed in 960 μL of a water/isopropanol mixed solvent (1:1 volume ratio) to form a homogeneous ink after 1 h of sonication. Then 16 μL catalyst ink was dropped onto the surface of a carbon cloth (0.4 × 0.4 cm$^2$), resulting in the loading of Pt/C being 0.5 mg/cm$^2$ (0.10 mg(Pt)/cm$^2$). For the 100 h stability test, the working electrodes were fabricated as follows: 5 mg of the catalyst Pt/C$_{60}$-2 and 2.5 mg ketjen black with 60 μL of Nafion solution were dispersed in 1440 μL of a water/isopropanol mixed solvent (1:1 volume ratio) over 1 h of sonication to form a homogeneous ink. Then 150 μL catalyst ink was dropped onto the surface of a carbon paper (1 × 1 cm$^2$), thus the loading of Pt/C$_{60}$-2 was 0.5 mg/cm$^2$ (0.108 mg(Pt)/cm$^2$).

The electrochemical measurements were carried out with a CHI 760E workstation using a conventional three-electrode cell in 1 M KOH at room temperature. A sample-coated rotating disk glass carbon electrode was the working electrode, a graphite rod was the counter electrode, and a saturated calomel electrode (SCE) was the reference electrode. In all measurements, the SCE reference electrode was calibrated with respect to the reversible hydrogen electrode (RHE). In N$_2$-saturated 1 M KOH, $E$(RHE) = $E$(SCE) + 1.0694 (1).

LSV measurements were conducted in 1 M KOH electrolyte at a scan rate of 5 mV s$^{-1}$. Cyclic voltammetry was conducted at a scan rate of 50 mV s$^{-1}$. In order to evaluate the reaction kinetics, the LSV curves

were redrawn as voltage and log current density to obtain a Tafel diagram. By fitting the linear part of the Tafel graph to the Tafel equation $\eta = a + b \log j$ (2) ($\eta$ is the overpotential, $b$ is the Tafel slope, and $a$ is the intercept), the Tafel slopes $b$ were obtained.

All the materials were subjected to electrochemical impedance spectroscopy (EIS) at open circuit potential with a frequency range between 0.1 Hz ~ 100 kHz and an amplitude of 5 mV. Cyclic voltammetry (CV) curves were recorded in the non-Faradic region with scanning rates of 10, 20, 30, 40, and 50 mV s$^{-1}$ to obtain the specific double-layer capacitance ($C_{dl}$) data by plotting the current difference ($\Delta j$) against the scanning rate. The exchange current densities ($J_0$) were determined from the intersection of the extrapolated linear part of the Tafel chart and the X-axis. The stability of the catalysts was tested by comparing the LSV curves before and after 3000 cycles of CV measurements. The long-term stability was performed through the chronopotentiometric test at 10 mA cm$^{-2}$.

The turnover of frequency (TOF) was calculated through the characterization of catalyst surface area via CO-stripping[50]. The CV baseline was first measured at a scan rate of 10 mV s$^{-1}$ in the N$_2$-saturated 1.0 M KOH electrolyte. Then a reduction potential (0.05 V vs. RHE) was applied to the working electrode, during which CO was bubbled into the electrolyte for 15 min to completely trigger CO adsorption on the catalyst, followed by the passage of N$_2$ for 15 min to allow the escape of CO from the electrolyte. CO was then oxidized through the CV scans, showing a distinct oxidation peak. The number of active sites exposed in the catalyst was determined by integrating the area of the oxidation peak. Then the TOFs were calculated by using the following equation:

$$TOF = j \cdot N_A / n \cdot F \cdot \Gamma \quad (3)$$

where the $j$, $N_A$, $n$, $F$, and $\Gamma$ are current density, Avogadro number ($6.023 \times 10^{23}$ mol$^{-1}$), number of electrons transferred ($n = 2$), Faraday constant (96485 C mol$^{-1}$), and the number of active sites, respectively.

Faraday efficiency measurement: Faraday efficiency (FE) for HER was determined by quantifying the volume of the generated H$_2$ gas through the drainage gas collection method. The FE was calculated by using the following equation:

$$FE = (V_1 / V_m) / (I \cdot T \cdot e / n \cdot N_A) \quad (4)$$

where the $V_1$, $V_m$, $I$, $T$, $e$, $n$, $N_A$ are measured gas volumes, the molar volume of gas at ambient temperature and pressure (24.45 L mol$^{-1}$), current, time, electric charge ($6.241 \times 10^{18}$ C$^{-1}$), number of electrons transferred ($n = 2$), and Avogadro number ($6.023 \times 10^{23}$ mol$^{-1}$), respectively.

The pH value of the prepared 1 M KOH electrolyte is 14.0, which was measured by a FE28-Standard pH meter. The specific operation is as follows: Firstly, calibrate the instrument with standard buffer solutions with pH values of 9.21 and 7.00, respectively. Then thoroughly clean the glass electrode with deionized water and dry it. Finally, insert the electrode into a 1 M KOH electrolyte and read the pH value of the electrolyte.

### Theoretical calculations

All density functional theory (DFT) calculations were implemented by the Vienna Ab initio Simulation Package (VASP 5.4.4)[61,62]. The electron exchange and correlation energies were handled using the generalized gradient approximation (GGA)[63] within the revised Perdew-Burke-Ernzerhof (RPBE) functionals[64]. The interactions between the cores and valence electrons were performed by the projector augmented wave (PAW) potentials[65]. The polarizable implicit solvent models were considered and the dielectric constant was set to 78.4 as implemented in the VASPsol[66]. In addition, the dispersion-corrected DFT-D3 scheme[67] was used to describe the van der Waals (vdW)

interactions. The cell parameters for Pt$_{13}$, C$_{60}$-Pt-C$_{60}$, and C$_{60}$-Pt$_{13}$-C$_{60}$ structure were set as a = 12 Å, b = 25 Å, c = 12 Å. The Monkhorst-Pack Γ-centered grids of 1 × 1 × 1 for all structures. A vacuum space 15 Å was employed to eliminate the interaction between the neighboring layers. The structure relaxations were carried out with a 500 eV plane-wave cutoff energy. The convergence criterion of structure optimization for energy and force were set as 10$^{-5}$ eV and 0.02 eV Å$^{-1}$, respectively.

## Data availability

The data that support the findings of this study are available from the Supplementary Information and/or from the corresponding author upon reasonable request.

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

## Acknowledgements

We appreciate the financial support from the National Key Research and Development Program of China (2021YFB3500700 to S.S.), the National Natural Science Foundation of China (22071070 to F.F.L., 21971077 to P.P., and 22025506 to S.S.). The US National Science Foundation (NSF) (CHE-1801317 to L.E.), and the Robert A. Welch Foundation for an endowed chair to L.E. (grant AH-0033 to L.E.) are also gratefully acknowledged.

## Author contributions

F.-F.L. and P.P. conceived the project and idea. R.Z., Y.L., and P.P. designed the experiments, R.Z. and Y.L. conducted the experiments and analyzed the data from these experiments, and wrote the manuscript. X.Z. and S.S. performed the atomic-resolution HAADF-STEM and XAS measurements. AY, Q.H., and T.X. participated in the sample synthesis and the electrochemical measurements. L.Z. performed DFT calculations. F.-F.L. edited the manuscript. L.E. edited and commented on the manuscript.

## Competing interests

The authors declare no competing interests.
