## [Peer review file · Nature Communications]

REVIEWER COMMENTS

Reviewer #1 (Remarks to the Author):

Comments to the contribution entitled: "Single-atomic platinum on fullerene C60 surfaces for accelerated alkaline hydrogen evolution"

Electrocatalytically initiated water splitting, i.e. water electrolysis when sourced by green electricity represents a promising route for the generation of hydrogen. The anodic half cell reaction is the main source for overpotential, however in particular when water electrolysis is aimed in alkaline medium the cathodic half cell reaction (hydrogen evolution reaction) presents a hurdle as well due to sluggish HER kinetics.

The authors suggest to use single-atomic platinum anchored on fullerene C60 as hydrogen evolution electrocatalyst. The electron withdrawing effect of the olefinic structure should lead to stabilized carbon-metal (covalent) bonds and to immobilized Pt atoms and hence to a reduction of the formation of bigger metal clusters. The C 60 approach was therefore used in order to reach a high loading of atomic Pt. the authors claim that the Pt loading amounts to 21% with a majority of single bonded Pt.

The preparation of Pt/C60 is not new and was shown by Nagashima et al. (Ref. 41 in the contribution). This synthetic approach was indeed used by the authors as well. Metallofullerenes

However, as far as I know Pt-C60 with single atomic Pt has not been used to support HER.

The paper is well written and the catalytic activity is convincing (25 mV overpotential for $j=10$ mA/cm²). However, the work seems to be incomplete and in its current stage not suitable for publication in a high ranked journal like Nat. Commun.

Revision required, subject to major revision.

1) Please insert page numbers!

2) (M@C60) with non PGM elements (M = Na, K, Rb, Cs, Sc, Ti, Mn, Fe) have been discussed as possible cathode materials for water electrolysis (Journal of Catalysis 354 (2017) 231–235).

This work should be cited here.

3) A strategy to achieve Pt containing HER electrode materials with minimal Pt losses during preparation has recently been shown and should be cited here (Angew. Chem. Int. Ed.2019,58, 17383 – 17392)

4. Water electrolysis was recently reviewed by Chatenet et al. (Chem. Soc. Rev., 2022, 51, 4583-4762). This is worth to be cited.

5 The authors used 0.1 M KOH and 1 M KOH (Fig S5) as electrolyte. This diluted KOH lacks of practicability and the authors need to show activity and stability of their material in e.g. 5 M KOH at elevated temperature (60 °C).

6) The electrocatalytic HER efficiency was determined on the basis of electric parameters (Figure 5). However, hydrogen gas was not quantified. The faradaic (HER) efficiency need to be determined, i.e. the charge to hydrogen conversion rate (Faradaic performance) needs to be experimentally determined.

7) Long term performance need to be shown ($t > 100000$ s) and the Pt loss need to be determined.

Reviewer #2 (Remarks to the Author):

This work demonstrates a simple wet-chemistry method to synthesise a single-atom Pt catalyst that performs well at alkaline HER, including high stability. This is therefore an interesting and potentially important result, however it is not a fundamental leap forward in this field but the improvement of single-atom catalysis. A very good wide range of analytical techniques used in this study, giving a range of information on bonding, structure/morphology, electronic/catalytic properties, also including DFT modelling. As highlighted in the comments, authors' claims are not entirely supported by these techniques, in particular the microscopy section where more information about the structure of the catalyst (e.g. single atom vs cluster, wt loading of Pt) could be extracted. The purpose of materials which have an excess of C60, as weight loading indicates any excess C60 has been removed, is not clear.

Detailed comments:

Line 23: Throughout the text there is some inconsistency with how the nature of the catalyst is described. Sometimes it is considered at a high loading of Pt on a C60 support (Pt/C60), whereas at other points it is described as a polymeric system of alternating C60 and Pt atom units (as in the source paper). There may well be advantages to considering the material as both a traditional catalyst and a metal coordinated polymer with covalent bonds, but if so a discussion of why both are useful is probably warranted for consistency.

Line 24: The nomenclature for the formula of the catalyst is inconsistent with literature. Chemistry Letters, (1994), 1207-1210. In this paper it is denoted C60Pt_n, indicating the nature of bonding between C60 and Pt to form a polymer. Pt/C60 implies a different morphology, and -2 is not clear that it refers to the stoichiometry.

Line 34: Terminology a bit vague. 'Cleanliness' to the environment?

Line 56: should read 'results'?

Line 76: 'perfect' as in spherical and so equal in any direction across the surface? Note that C60 is a polyhedron, not a sphere.

Line 82: Geometrically perhaps, but steric hinderance and C60 cage distortion limit this to about 6 at most.

Lines 89-90: The source paper (Chemistry Letters, (1994), 1207-1210) synthesise C60Pt_n and test it's catalytic performance for the hydrogenation of diphenylacetylene. This statement is therefore somewhat misleading.

Lines 110-111: Why were these different ratios chosen? The source paper discusses 1:1 molar ratios of C60:Pt, and attempted to synthesise higher Pt ratios. For 1:1 C60:Pt and higher C60 ratios, as in this paper, what was the target material? 1:1 should yield a 1-dimensional polymer of alternating C60 and single Pt atoms (as shown in Fig. 1), so what is the expected product with x2 and x4 the quantity of C60? It is unlikely more than 2/3 C60 molecules could bind to a single Pt atom, and so the washing step in the synthesis will likely wash any excess C60 away, yielding a similar product regardless of what ratio of C60:Pt was used. Explanation of this choice is necessary.

Lines 116-117: Again, this nomenclature isn't consistent or immediately obvious to what it is referring.

Lines 118-123: of course, these analyses confirm there is no presence of dba, and so the only remaining components are C60 and Pt (and possibly residual solvent). Is it possible that much of the product is a mixture of separated aggregates of both C60 and Pt?

Lines 123-125: This is the first mention of "fulleride", i.e. C60²⁻ Pt²⁺. A discussion of the oxidation state of Pt and the charge of C60 in C60Pt_n would be useful in the subsequent analysis. Is there agreement with Raman for the 'fulleride' state of C60?

Line 125: Again, inconsistent nomenclature. The original paper refers to C60Pt_n where n is the ratio of Pt to C60 in the material. Here [C60-Pt]_n, n refers to the chain length of repeating C60-Pt units.

Line 134: Why is the catalyst with 2:1 ratio of C60:Pt the main material of study? In the figure and text above it is agreed a linear polymer of alternating C60 and Pt unit is the target material (with C60:Pt 1:1). Where is the 100% increase in C60 utilised? What is the purpose in the catalyst, and was any excess C60 removed during the washing step?

Line 135: 'PXRD' - maybe use this acronym to avoid confusion with single crystal XRD.

Line 160: 'species'.

Line 162: How does this compare to the theoretical Pt mass loading?

For Pt/C60-1 = 21.3 wt% Pt

For Pt/C60-2 = 11.9 wt% Pt

This strongly suggests that the C60-Pt polymer was formed with 1:1 C60:Pt, hence any excess C60 was removed. Again, what was the purpose of using Pt/C60-2 being the main material of study? (over Pt/C60-1. Note, in practice, the quoted loadings make little sense because the catalyst is diluted with ketjen black to improve conductivity, hence loadings are lower in reality.

Line 177: 'large number' is quite vague. Any attempt at quantisation would be useful (atoms per nm²?)

Line 177: this is the biggest problem here. The STEM images here and in the SI clearly show several crystalline and amorphous Pt particles, which will contain the vast majority of Pt atoms.

Line 178: Much too high accuracy in measurement. At 200 kV the electron wavelength is about 3 pm, and the pixel size is about 25 pm, which will be the limiting factor in resolution. Quoting distance measurements to 1 pm is nonsense here. Quoting to two decimal places is more appropriate (0.28 nm) In the SI there are 4 d-spacing measurements made all of which are quoted as exactly 0.227 nm, which does not seem plausible, even if the software is quoting to 3 decimal places.

Line 180: Are there? What is the proof that they are single atoms and not clusters?

The expected morphology of the catalyst is a 1D chain. Are there areas that show chains of Pt atoms. What is the expected separation of these based on models and how close are these to reality?

Line 182: What proportion of surface Pt atoms would this particle size have? How would this impact on the activity of the catalyst?

Line 183: images look like STEM-EDX, if so please mention the correct technique. The choice of area and magnification for the EDX are odd, the images only show that C and Pt are both present fairly homogeneously. The high contrast areas in the HAADF probably correspond to Pt clusters, supported by STEM-EDX, further suggesting there are areas of agglomerated Pt. Additionally, what are the measured abundances of C and Pt from the EDX, and how do these compare to both experimental and theoretical Pt weight loading?

Lines 186-187: I'm not suggesting that areas of high contrast are all Pt clusters and not clusters of single atoms, it may well be the case that is suggested in the text. However, these TEM images are not sufficient to support that claim, indeed I believe they evidence the opposite case that the number of clusters greatly outweighs the number of single atoms.

Fig3b: What does this image show? An amorphous particle edge, but otherwise not much useful information here. How were the TEM grids prepared? This would be useful to know in order to see what the background in this image is.

Fig3: STEM images - as above, there are many particles and agglomerations of Pt atoms. How big are these and what is the expected proportion of surface atoms. Additionally, what is the total expected ratio of single atoms to clusters

Line 207: Coordination number is measured to be ~ 3 , why is there a discrepancy if the coordination is Pt-C4? A mixture of products or another product entirely? Maybe discuss why this difference could be present.

Line 210: Again, STEM results need more analysis to fully support this claim, at the moment STEM images appear to claim the opposite.

Electrochemistry section:

a) why there is no discussion of LSV? The catalyst exhibits no features associated with HER (Fig.S4). Why?

b) Electrocatalyst ink preparation procedure (sonication of Pt-C60 compounds with nafion and ketjen in water/IPA for several hours) is very aggressive and may completely change the nature and structure of the catalyst. There is no guarantee that single atoms survive this process.

c) Note that C60 reacts readily with KOH forming fullerols, so the catalyst will undergo further chemical changes when immersed in the electrolyte, i.e. it will not be the same material as it is believed and modelled in the paper.

Line 290: Again, why were these ratios chosen? Nomenclature of materials is inconsistent

Lines 291-292: Choice of TEM images in Fig 6b are poor to illustrate this point. It is not very clear what areas correspond to Pt clusters, and indeed why they are clusters here but not in the STEM imaging in Figure 3. Higher magnification imaging over a wider area would be useful. The second image of Fig 6b is not illustrative of anything other than amorphous material, which could be a range of things. EDX analysis would be useful again, both STEM mapping and qualitative abundances of C and Pt.

Lines 314-316: What about the Pt atom in the centre of the cluster? Should this say there are two types of 'surface' Pt atoms?

Fig 7b: Pt₁₃ cluster is bonded across the hexagonal face of C60? Why was this bonding mode chosen?

Line 348: For single Pt atom catalysis, adsorption of H₂O and H both have positive ΔG . Is positive ΔG for H₂O adsorption considered optimal?

Lines 356-358: From Fig. 7f it is difficult to qualitatively assess the area under each curve in the blue rectangle. To my eye, I would say that there is a decrease from red to blue, then an increase from blue to yellow. Quantifying the area under each curve may be useful to support the claim of the text.

Line 374: are these definitely fullerides? What is evidence for C60 being anion?

Line 390: ¹H NMR? - please mention specific nucleus.

Line 399: Observation - this is the exact same description as from the original Nagashima paper.

Materials characterisation section:

EDX - how was this recorded? (described as 'TEM mapping' in this text).

Lines 464-466: Again, evidence may be that not all Pt sites are exposed. From the average particle size data from TEM, estimating how the surface:bulk Pt atomic ratio assuming there are no single atoms will give a lower bound on the activity. How does this value compare to other literature and other experimental results in this text? Maybe estimation of the single-atom : particle is possible via this method - I think the single biggest unanswered question here.

Line 480: should read 'structures'?

Reference 42: This is the wrong year of publication for this journal. It should be 1994.

Reviewer #3 (Remarks to the Author):

Single-atomic platinum on fullerene C60 surfaces for accelerated alkaline hydrogen evolution by Zhang, et al. deals with the synthesis and characterization of polymeric PtC60 species and their application as catalyst for the HER. The paper is well planned and executed, and clearly written, giving key literature. The authors claim that the PtC60 species outperforms the benchmark 20 wt% Pt/C. For me this is a key point of the paper, and I think that the authors do not compare both catalysts with the exact Pt loading, if I well understood. If the electrode was prepared that way : 5mg of the catalyst and 2.5 mg ketjen black with 60 μ L of nafion solution ; for all catalyst, the authors should consider the Pt available for catalysis, as the catalysts are very different in nature. That is, PtC60 should be less loaded than that of 20%Pt/C, as I assume the Pt is in nanoparticle form for the later (more information about the 20%Pt/C should be given in the manuscript). Maybe a chemisorption of both catalysts should be given and the catalytic tests corrected to this data. If authors consider this point, and it is confirmed that PtC60 outperforms Pt over carbon, the paper should be worthy to publish to this journal.

Apart from the chemisorption to obtain the Pt available for catalysis, other points should be revised:

It is true that PtC60 species have been prepared in the past, among others, but poorly characterized, I appreciate the effort that has been done in this sense. Nevertheless, the authors claim: C60 engineered

single-atomic metal catalysts and their catalytic performances have never been reported; but they should cite: 10.1039/d0cy00540a; 10.1021/acscatal.6b01429; 10.1039/c9cy02025j

They also claim: Therefore, we can conclude that Pt atoms bind to C60 molecules to form a one-dimensional metal-fulleride polymer with the chemical formula $[C60-Pt]_n$ ($n > 1$), in which Pt and C60 are alternately arranged.⁴²; I have difficulties to understand this conclusion based in the Pt content.

In the DFT calculations part the authors compare adsorption energies of H₂O and H on the surface of the single-atom and Pt₁₃ coordinated to fullerene with that of Pt(111). It is known that the adsorption energies can vary from flat surfaces to nanoparticles, it will be more accurate to compare with a bare Pt₁₃ as well.

Reviewer #4 (Remarks to the Author):

The manuscript by Zhang et al. discusses a novel organometallic Pt(C60) oligomer catalyst for H₂ evolution reaction. In the first part of the manuscript they describe the synthesis of the material and carefully characterize it using a variety of X-ray techniques (XPS, XANES), IR and Raman spectroscopy and microscopy. Based on this analysis they conclude that they successfully synthesized the target complex. This analysis is very detailed and the used techniques definitely support this conclusion. The obtained catalyst is then characterized by electrochemical techniques (voltammetry and impedance spectroscopy) based on which they conclude that the obtained material is slightly more active than pure Pt(111). This is then followed by an analysis of the potential active site using catalysts with decreasing amount of Pt nanoparticles and through poisoning experiments. Here the authors convincingly show that the Pt(C60)₂ complex is indeed the active species for H₂ evolution. These results are then rationalized by DFT modelling to obtain a detailed understanding of the underlying mechanisms. This section is unfortunately not fully convincing.

Overall this is a very careful and interesting study which has been summarized in a well written manuscript. I can support its publication after a few minor questions have been clarified.

Minor issues:

1) p.2: In the introduction the authors claim that already today electrolysis stands out for its simplicity and the availability of catalytic materials. This is unfortunately not yet true. The HER requires scarce and

expensive Pt as a catalyst which can not yet be replaced easily. In line with this, the main technique to obtain H₂ in industry is steam reforming of natural gas.

II) p.2: In the introduction the authors claim that the HER kinetics are hindered by oxygen intermediates. This sounds a bit strange. I would expect, that water reduction results in the formation of adsorbed *H and a dissolved OH⁻ not the formation of adsorbed *OH or *O groups. It would be good if this issue is clarified by the authors.

III) line 249: Impedance spectroscopy indicates a lower charge transfer resistance for the Pt(C60) complexes. Why do the authors believe that this is not the sole reason for the very minor improvement in the performance?

IV) line 294: The authors state here that the Pt/C60-2 catalyst solely contains Pt(C60). This is inconsistent with earlier parts of the manuscript where they show that at least minor amounts of Pt nanoparticles are present.

V) line 311: The model description is not completely clear. Do the authors use a periodic model of the Pt(C60) complex or does it consist of a single isolated Pt(C60)₂ monomer? If the later is true why did they choose a plane wave code with pbc rather than one of the standard molecular codes which would give access to better DFT functionals and more advanced solvation models? For the periodic model also the box size and the amount of vacuum between the layers must be described in the computational details.

VI) Figure 7: The mechanism suggested by the authors which proceeds through the adsorption of H₂O followed by the formation of an H adsorbate is not fully convincing. They argue that owing to a thermodynamic barrier for H₂O adsorption Pt(C60) is a better catalyst than Pt(111). On the other hand the authors show, that the adsorption energy of *H (which is used as descriptor) is with -160 mV for Pt(111) much closer to the ideal than any of the Pt(C60) complexes. Thus, I would expect that Pt(111) is the best catalyst. Disregarding Pt(111) on the other hand based on the H₂O adsorption energy is in my opinion not fully correct since H adsorption could also occur through an Eley-Rideal type mechanism which does not require the H₂O to pre-adsorb at the surface.

VII) Figure 7: The difference in the H-adsorption energy between Pt(111) and Pt-C60-1 is with 140 mV rather minor. How large is the error the authors expect for their organometallic catalyst and how robust do they expect the trends to be with respect to the choice of functional/method (e.g. a meta-GGA or GGA+U)?

VIII) Figure 7: The adsorption energy of water on C60-Pt13 is with -1.43 eV extremely negative. This is somewhat surprising since I would expect that water only physisorbes extremely weakly. Do the authors have any explanation for this surprising result?

February 9, 2023

Dear Reviewers,

We would like to thank all the reviewers for your time in reviewing our manuscript entitled “Single-atomic platinum on fullerene C₆₀ surfaces for accelerated alkaline hydrogen evolution” (Manuscript ID: NCOMMS-22-30291-T). We highly appreciate the valuable and constructive comments. The critical comments and kind suggestions are important to improve the manuscript. As suggested, the manuscript and the supplementary information have been revised with the addition of more information and all the changes are highlighted in yellow in the revised manuscript. Below, we present our point-by-point response (in blue) to address all the questions, comments, and issues raised by you (in black) in details. We hope that the revised manuscript can meet the demands of you.

With best regards,

Fang-Fang Li, Ph.D.

Email: ffli@hust.edu.cn

General changes:

- The first author Ruiling Zhang graduated in July 2022, the second author Yaozhou Li did the experiments and revised the manuscript, therefore, Yaozhou Li is added as the co-first author in the revised manuscript. Qi Huang performed the CO-stripping experiments for TOF calculations, he is added as one of the co-authors.
- According to the “Formatting instructions”, a short, standalone title was added to each figure.
- According to the “Formatting instructions”, references should be limited to 70, therefore, the previous references 1, 5, 12, 18, 25, 26, 27, 29, 35, 50, 52, 53, 57, 67, and 69 have been removed and the total reference number is now 68 in the revised manuscript. The removed references are mainly located where there are multiple citations and thus do not affect the understanding of the manuscript.

Response to Reviewers

We are very grateful to the reviewers for the valuable and constructive comments, which helped us improve the manuscript and make a much better presentation of the work.

Reviewer #1 (Remarks to the Author):

Comments to the contribution entitled: “Single-atomic platinum on fullerene C60 surfaces for accelerated alkaline hydrogen evolution”

Electrocatalytically initiated water splitting, i.e. water electrolysis when sourced by green electricity represents a promising route for the generation of hydrogen. The anodic half cell reaction is the main source for overpotential, however in particular when water electrolysis is aimed in alkaline medium the cathodic half cell reaction (hydrogen evolution reaction) presents a hurdle as well due to sluggish HER kinetics.

The authors suggest to use single-atomic platinum anchored on fullerene C60 as hydrogen evolution electrocatalyst. The electron withdrawing effect of the olefinic structure should lead to stabilized carbon-metal (covalent) bonds and to immobilized Pt atoms and hence to a reduction of the formation of bigger metal clusters. The C 60 approach was therefore used in order to reach a high loading of atomic Pt. the authors claim that the Pt loading amounts to 21% with a majority of single bonded Pt.

The preparation of Pt/C60 is not new and was shown by Nagashima et al. (Ref. 41 in the contribution). This synthetic approach was indeed used by the authors as well. Metallofullerenes However, as far as I know Pt-C60 with single atomic Pt has not been used to support HER.

The paper is well written and the catalytic activity is convincing (25 mV overpotential for $j=10$ mA/cm²). However, the work seems to be incomplete and in its current stage not suitable for publication in a high ranked journal like Nat. Commun.

Revision required, subject to major revision.

- We highly appreciate your valuable comments and suggestions on this work. Please see below our point-by-point responses to your concerns.

1. Please insert page numbers!
 - Page numbers have been inserted in the revised manuscript.

2. (M@C₆₀) with non PGM elements (M = Na, K, Rb, Cs, Sc, Ti, Mn, Fe) have been discussed as possible cathode materials for water electrolysis (Journal of Catalysis 354 (2017) 231–235). This work should be cited here.
 - Many thanks for informing us of this informative work, which has been cited as reference 31. The following comments have been added to **Introduction** of the revised manuscript. “Besides, theoretical studies have demonstrated that charge transfer from metal to C₆₀ cage can enhance the adsorption of H on endohedral metallofullerenes (M@C₆₀) during hydrogen evolution”.

3. A strategy to achieve Pt containing HER electrode materials with minimal Pt losses during preparation has recently been shown and should be cited here (Angew. Chem. Int. Ed.2019,58, 17383-17392)
 - This excellent work has been cited as reference 5 in the revised manuscript.

4. Water electrolysis was recently reviewed by Chatenet et al. (Chem. Soc. Rev., 2022, 51, 4583-4762). This is worth to be cited.
 - This comprehensive review has been cited as reference 2 in the revised manuscript.

5. The authors used 0.1 M KOH and 1 M KOH (Fig S5) as electrolyte. This diluted KOH lacks of practicability and the authors need to show activity and stability of their material in e.g. 5 M KOH at elevated temperature (60 °C).
 - As suggested, both Pt/C₆₀-2 and Pt/C were tested in 5 M KOH at 60 °C. The working electrodes were fabricated as follows: 5mg of the catalyst Pt/C₆₀-2 and 2.5 mg ketjen black with 60 μL of Nafion solution were dispersed in 1440 μL of a water/isopropanol mixed solvent (1:1 volume ratio) over 1h hour of sonication to form a homogeneous ink. Then 24 μL catalyst ink was dropped onto the surface of a carbon cloth (0.4×0.4 cm²), thus the loading of Pt/C₆₀-2 was 0.5 mg/cm². For the Pt/C electrode, 5mg of Pt/C with 40 μL Nafion solution were dispersed in 960 μL of a water/isopropanol mixed solvent (1:1 volume ratio) to form a homogeneous ink after 1h hour of sonication. Then 16 μL catalyst ink was dropped onto the surface of a carbon cloth (0.4×0.4 cm²), resulting in a loading of Pt/C of 0.5 mg/cm². As shown in Fig. R1, Pt/C₆₀-2 shows a lower overpotential ($\eta_{10} = 26.1$ mV) than Pt/C ($\eta_{10} = 32.1$ mV) at the current density of 10 mA cm⁻². Moreover, after 3000 cyclic CV tests, the overpotential of Pt/C₆₀-2/KB at 20 mA cm⁻² increased only 4 mV, whereas a 23 mV increase is observed for Pt/C, indicating that Pt/C₆₀-2 still exhibits superior activity and stability over Pt/C under harsh conditions. The relevant discussion has been added to the revised manuscript, and Fig. R1b has been added as Fig. 5i of the revised manuscript.

Fig. R1. (a) HER polarization curves for Pt/C₆₀-2/KB and 20 wt% Pt/C in 5 M KOH at 60 °C. (b) LSV curves for Pt/C₆₀-2/KB and 20 wt% Pt/C before and after 3000 catalytic cycles in 5 M KOH at 60°C.

6. The electrocatalytic HER efficiency was determined on the basis of electric parameters (Figure 5). However, hydrogen gas was not quantified. The faradaic (HER) efficiency need to be determined, i.e. the charge to hydrogen conversion rate (Faradaic performance) needs to be experimentally determined.

■ The Faradaic efficiency of Pt/C₆₀-2 for HER was determined by quantifying the volume of generated H₂ through the drainage gas collection method. The results showed that the Faradaic efficiency is above 98 % over 4h. The relevant information has been added to the manuscript and Fig. R2 below has been added to the revised Supplementary information as Supplementary Fig. 11

Fig. R2. (a) Theoretical and experimentally collected H₂ volumes. (b) Faraday efficiency for Pt/C₆₀-2.

7. Long term performance needs to be shown ($t > 100000$ s) and the Pt loss need to be determined.

■ The long-term stability of 100 h at 10 mA cm⁻² in 1 M KOH was monitored, and the overpotential only increased 31.5 mV, much better than that for Pt/C (Fig. R3). TEM images showed that the spherical morphology of Pt/C₆₀-2 was retained after the stability test (Fig. R4a-d). Moreover, Pt single atoms are clearly observed in the HAADF-STEM images (Fig. R4e),

indicating that Pt single atoms are very stable during catalysis. Consistent results were obtained from XPS, since the peak positions did not change relative to those before the catalysis (Fig. R4f). Although more Pt clusters were formed, the dimensions of the clusters remained unchanged during the long-term stability test (Fig. R4c-e). To determine Pt leaching after the stability test, Inductively coupled plasma-mass spectrometry (ICP-MS) was conducted to analyze the electrolyte, and the concentration of Pt ions was determined to be only 0.034 mg L^{-1} , further indicating the robust nature of Pt/C₆₀-2.

- The above discussion has been added to the revised manuscript. Fig. R3 was added as Fig. 5g of the revised manuscript. Fig. R4 was added as Supplementary Fig. 10 of the revised Supplementary information.

Fig. R3. Long-term stability test at a current density of 10 mA cm^{-2} in 1 M KOH.

Fig. R4. TEM images of Pt/C₆₀-2 before (a,c) and after (b,d) 100 h long-term stability test. (e) HAADF-STEM image of the catalyst after 100 h durability test. (f) XPS spectra of the catalyst before and after the durability test.

Reviewer #2 (Remarks to the Author):

This work demonstrates a simple wet-chemistry method to synthesise a single-atom Pt catalyst that performs well at alkaline HER, including high stability. This is therefore an interesting and potentially important result, however it is not a fundamental leap forward in this field but the improvement of single-atom catalysis. A very good wide range of analytical techniques used in this study, giving a range of information on bonding, structure/morphology, electronic/catalytic properties, also including DFT modelling. As highlighted in the comments, authors' claims are not entirely supported by these techniques, in particular the microscopy section where more information about the structure of the catalyst (e.g. single atom vs cluster, wt loading of Pt) could be extracted. The purpose of materials which have an excess of C₆₀, as weight loading indicates any excess C₆₀ has been removed, is not clear.

Detailed comments:

- We highly appreciate your valuable comments and suggestions on this work. Please see below our point-by-point responses to your concerns.
1. Line 23: Throughout the text there is some inconsistency with how the nature of the catalyst is described. Sometimes it is considered as a high loading of Pt on a C₆₀ support (Pt/C₆₀), whereas at other points it is described as a polymeric system of alternating C₆₀ and Pt atom units (as in the source paper). There may well be advantages to considering the material as both a traditional catalyst and a metal coordinated polymer with covalent bonds, but if so a discussion of why both are useful is probably warranted for consistency.
- In the reactions between C₆₀ and metal complexes, metal-C₆₀ polymers are usually formed when the ligands are completely replaced by C₆₀. The presence of a vibration at 437 cm⁻¹ in the IR spectrum (Fig. 2b) and the absence of C=O bonds in the XPS spectrum of Pt/C₆₀-2 (Fig. 2d) indicate the complete replacement of the two dba ligands by two C₆₀ molecules, suggesting the formation of Pt-C bonds as well as the polymeric structure. Therefore, Pt/C₆₀-2 is a high-loading Pt catalyst whose inner structure can be described as a platinum-C₆₀ polymeric structure. The A_g(2) mode of C₆₀ in the Raman spectrum is an indicator for chemical modifications of the C₆₀ cage surface (Fig. 2c). The 14 cm⁻¹ downshift of the A_g(2) band of Pt/C₆₀-2 relative to that of C₆₀ confirmed the cage modification by Pt. The poisoning experiments confirm the presence of both Pt single atoms and Pt clusters in Pt/C₆₀-2, whereas the intense peak corresponding to Pt-C coordination in the FT-EXAFS spectra of the Pt L₃-edge of Pt/C₆₀-2 (Fig. 4b) support the predominance of Pt single atoms. In contrast, the significantly weaker peak of Pt-Pt coordination suggests a smaller proportion of Pt clusters. The predominance of Pt single atoms was further confirmed by cyclic voltammetry (CV) measurements. The CV curve showed no features associated with H₂ adsorption-desorption due to the oxidation state of the metal species. As displayed in Supplementary Fig. 6 of the revised Supplementary information, unlike Pt/C, which is dominated by Pt nanoparticles (Supplementary Fig. 7), the CV curve of Pt/C₆₀-2 exhibits no hydrogen adsorption-desorption characteristic peaks, indicating that Pt atoms in Pt/C₆₀-2 are mainly atomically dispersed. In addition, since there are 4 Pt-C bonds for each Pt atom, a coordination number of 3 indicates that Pt clusters also connect to C₆₀ to lower the coordination

number. Based on the above analysis, the description on the structure of Pt/C₆₀-2 has been revised and unified as “Pt/C₆₀-2 is a high-loading Pt catalyst whose inner structure can be described as a platinum-C₆₀ polymeric structure, in which the predominant Pt single atoms serve as bridges connecting the adjacent C₆₀, and Pt clusters are anchored on C₆₀ as well”. It is worth noting that Pt/C₆₀-1, Pt/C₆₀-2, and Pt/C₆₀-4 are only the names of the synthesized catalysts, without specific structural information.

2. Line 24: The nomenclature for the formula of the catalyst is inconsistent with literature. *Chemistry Letters*, (1994), 1207-1210. In this paper it is denoted C₆₀Pt_n, indicating the nature of bonding between C₆₀ and Pt to form a polymer. Pt/C₆₀ implies a different morphology, and -2 is not clear that it refers to the stoichiometry.
 - Our nomenclature of the catalysts is indeed different from that in the cited literature. In *Chemistry Letters*, (1994), 1207-1210, the sample is denoted as C₆₀Pt_n, where n represents the atomic ratio of Pt to C₆₀ in the samples. The article claimed that C₆₀Pt_n has a 1D polymeric structure with alternating C₆₀ and Pt atoms. However, the structural characterizations of C₆₀Pt_n in the source paper are too limited to show the real structure of the sample. In fact, according to our studies, the structure of the sample contains both Pt single atoms and Pt clusters. Therefore, instead of using the same nomenclature for catalysts, we use Pt/C₆₀-1, Pt/C₆₀-2, and Pt/C₆₀-4, where 1, 2, and 4 represent the molar ratio of C₆₀ to Pt(dba)₂ during sample synthesis. The molar ratios of C₆₀ to Pt(dba)₂ were set to 1:1, 2:1, and 4:1, respectively, so we named the catalysts Pt/C₆₀-1, Pt/C₆₀-2, and Pt/C₆₀-4. It is worth noting that Pt/C₆₀-1, Pt/C₆₀-2, and Pt/C₆₀-4 are only the names of the synthesized samples, without specific structural information.
3. Line 34: Terminology a bit vague. ‘Cleanliness’ to the environment?
 - Thanks for the suggestion. “Cleanliness” has been corrected to “Cleanliness to the environment” in the revised manuscript.
4. Line 56: should read ‘results’?
 - Sorry for making this mistake. “result” has been corrected to “results” in the revised manuscript.
5. Line 76: ‘perfect’ as in spherical and so equal in any direction across the surface? Note that C₆₀ is a polyhedron, not a sphere.
 - Thanks for the suggestions. The terms “spherical” and “perfect” have been deleted in the revised manuscript. The updated statement is as follows: “Fullerenes are zero-dimensional nanocarbons with well-defined molecular structures.... Of these, C₆₀, with icosahedral (*I_h*) symmetry, not only possesses a curved surface but”
6. Line 82: Geometrically perhaps, but steric hinderance and C₆₀ cage distortion limit this to about 6 at most.
 - Thanks for the suggestions. “C₆₀ has 30 C=C bonds on its curved surface, which in theory could bind 30 metal atoms” has been revised to “C₆₀ has 30 C=C bonds on its curved surface, which in theory could bind multiple metal atoms” in the manuscript.

7. Lines 89-90: The source paper (Chemistry Letters, (1994), 1207-1210) synthesize $C_{60}Pt_n$ and test its catalytic performance for the hydrogenation of diphenylacetylene. This statement is therefore somewhat misleading.
- “ C_{60} engineered single-atomic metal catalysts and their catalytic performances have never been reported” has been revised to “However, the application of C_{60} engineered single-atomic metal catalysts in the field of electrocatalysis and their experimental catalytic performances have never been explored.” in the revised manuscript.
8. Lines 110-111: Why were these different ratios chosen? The source paper discusses 1:1 molar ratios of C_{60} : Pt, and attempted to synthesize higher Pt ratios. For 1:1 C_{60} : Pt and higher C_{60} ratios, as in this paper, what was the target material? 1:1 should yield a 1-dimensional polymer of alternating C_{60} and single Pt atoms (as shown in Fig. 1), so what is the expected product with x2 and x4 the quantity of C_{60} ? It is unlikely more than 2/3 C_{60} molecules could bind to a single Pt atom, and so the washing step in the synthesis will likely wash any excess C_{60} away, yielding a similar product regardless of what ratio of C_{60} : Pt was used. Explanation of this choice is necessary.
- Yes, the source paper mainly discussed the 1:1 molar ratio of C_{60} : Pt, and proposed the structure of the synthesized $C_{60}Pt_n$ a one-dimensional chain model of alternating Pt atoms and C_{60} molecules. We initially synthesized the catalyst by following the conditions reported in the source paper, i.e. the molar ratio of C_{60} : Pt is 1:1. However, the structure characterizations showed that there are many Pt clusters formed in the Pt/ C_{60} -1 sample (Fig. 6b,d and Fig. R5). In order to reduce the content of Pt clusters, we increased the amount of C_{60} , i.e. the molar ratio of C_{60} : Pt increased to 2:1 and 4:1, to better disperse the Pt during the synthesis process and prevent the aggregation of Pt. HR-TEM and XPS results of Pt/ C_{60} -1/2/4 (Fig. 6b,d and Fig. R5) revealed that with the increase of C_{60} , the aggregation of Pt in Pt/ C_{60} -2/4 is decreased. Although the excess C_{60} is washed away by toluene, it plays a role in inhibiting Pt aggregation during sample synthesis. Pt/ C_{60} -1, Pt/ C_{60} -2, and Pt/ C_{60} -4 are similar products, but the relative contents of Pt clusters are different. Fig. R5 has been added as Supplementary Fig. 12 of the revised Supplementary information.

Fig. R5. TEM images of (a) Pt/ C_{60} -1 and (b) Pt/ C_{60} -4 over a wider area, showing more abundant Pt clusters in Pt/ C_{60} -1.

9. Lines 116-117: Again, this nomenclature isn't consistent or immediately obvious to what it is referring.
- Please see response to reviewer #2 point #2 on page #7.

10. Lines 118-123: of course, these analyses confirm there is no presence of dba, and so the only remaining components are C₆₀ and Pt (and possibly residual solvent). Is it possible that much of the product is a mixture of separated aggregates of both C₆₀ and Pt?
- According to the structure characterizations for Pt/C₆₀-2, Pt and C₆₀ are connected by Pt-C covalent bonds, rather than a mixture of separated aggregates of both C₆₀ and Pt.
 - FT-IR spectrum of Pt/C₆₀-2 showed a red-shifted and intense vibration at 437 cm⁻¹ (Fig. 2b), which corresponds to the Pt-C bond, suggesting that C₆₀ and Pt are covalently bonded. The formation of Pt-C bonds in Pt/C₆₀-2 was further confirmed by Raman spectra (Fig. 2c). A_g(2) mode of C₆₀ in the Raman spectrum is an indicator for chemical modifications of the C₆₀ cage. The downshift of the A_g(2) band confirmed the cage modification by Pt and the oxidation state of Pt being Pt²⁺. In addition, the intense peak corresponding to Pt-C coordination in the FT-EXAFS spectra of the Pt L₃-edge of Pt/C₆₀-2 (Fig. 4b) further confirmed Pt-C bond formation between Pt and C₆₀. While the intensity of the Pt-Pt bond is quite small, suggesting that Pt single atoms are dominant in Pt/C₆₀-2. The Pt-C bond was further verified by EXAFS wavelet transform analysis (Fig. 4d-f) by showing an intense peak at 3.9 Å, which corresponds to the Pt-C bond. Whereas the Pt-Pt bond was not observed, indicative of less amounts of Pt clusters in the Pt/C₆₀-2 sample. Furthermore, cyclic voltammetry (CV) measurement of Pt/C₆₀-2 also indicated the atomically dispersed nature of Pt species by showing no features associated with H₂ adsorption-desorption (Supplementary Fig. 6). If the product is a mixture of separated aggregates of both C₆₀ and Pt, we should detect obvious hydrogen adsorption-desorption characteristic peaks like the peaks in the CV curve of Pt/C (Pt/C is dominated by Pt⁰ nanoparticles). Based on the above analyses, the product is unlikely to be a mixture of separated aggregates of both C₆₀ and Pt.
11. Lines 123-125: This is the first mention of "fulleride", i.e., C₆₀²⁻ Pt²⁺. A discussion of the oxidation state of Pt and the charge of C₆₀ in C₆₀Pt_n would be useful in the subsequent analysis. Is there agreement with Raman for the 'fulleride' state of C₆₀?
- The oxidation state of Pt has been probed by XPS (Fig. 2d) and XANES (Fig. 4a) measurements. The Pt 4f peaks of Pt/C₆₀-2 are positively shifted relative to those of Pt⁰(dba)₂, indicating that the oxidation state of the Pt atoms increased due to the electron transfer from Pt to C₆₀. Moreover, the binding energies of the two peaks (75.43 and 72.05 eV) correspond to the 4f_{5/2} and 4f_{7/2} orbitals of the Pt²⁺ species (Fig. 2d). The oxidation state of Pt in Pt/C₆₀-2 was further confirmed by the X-ray absorption near-edge structure (XANES) spectra of Pt L₃-edge (Fig. 4a), where the strong white line (WL) of Pt/C₆₀-2 suggests an elevated oxidation state for Pt, which agrees well with the XPS results. Besides, the oxidation state of Pt in Pt/C₆₀-2 was examined by Raman measurements (Fig. 2c). The A_g(2) mode of C₆₀ in the Raman spectrum is sensitive to chemical modifications and is commonly accepted as an indicator for charge transfer between the metal and C₆₀. As one electron is transferred to C₆₀, the A_g(2) mode is downshifted by ca. 6 cm⁻¹ (*Thin Solid Films* 2003, 429, 96-101; *Catalysis Science & Technology*, 2020, 10, 4673-4683, *RSC Adv*, 2016, 6, 69135). The A_g(2) peak for Pt/C₆₀-2 (1448 cm⁻¹) decreased by 14 cm⁻¹ relative to that of pure C₆₀ (1462 cm⁻¹) (see updated Fig. 2c), indicating ~2.3 electrons were transferred from Pt to C₆₀, forming the Pt-fulleride. The discussions on the Raman results have been added to the revised manuscript. Fig 2c has been updated with Fig. R6.

Fig. R6. Raman spectra of Pt/C₆₀-2 and C₆₀.

12. Line 125: Again, inconsistent nomenclature. The original paper refers to C₆₀Pt_n where n is the ratio of Pt to C₆₀ in the material. Here [C₆₀-Pt]_n, n refers to the chain length of repeating C₆₀-Pt units.
 - For inconsistent nomenclature, please see response to reviewer #2 point #2 on page #7. Based on current structural characteristics, we have revised the statement to “Pt/C₆₀ catalysts thus have the platinum-C₆₀ polymeric structures. And the Pt atoms are mainly in a divalent form (see Raman, XPS, and XAS results below), indicating the atomically dispersed Pt single atoms predominate and prefer to be tetracoordinate.” in the revised manuscript.

13. Line 134: Why is the catalyst with 2:1 ratio of C₆₀:Pt the main material of study? In the figure and text above it is agreed a linear polymer of alternating C₆₀ and Pt unit is the target material (with C₆₀:Pt 1:1). Where is the 100% increase in C₆₀ utilised? What is the purpose in the catalyst, and was any excess C₆₀ removed during the washing step?
 - HER catalytic activities of Pt/C₆₀-1, Pt/C₆₀-2, and Pt/C₆₀-4 were examined and the results showed that Pt/C₆₀-2 exhibited the best electrocatalytic hydrogen evolution performance, so Pt/C₆₀-2 was selected as the target material to analyze its structure, investigate its catalytic performance and the structure-activity relationship. The unreacted C₆₀ was washed off by toluene and can be recycled.
 - As for the increased C₆₀ during the sample synthesis, see response to reviewer #2 point #8 on page #8.

14. Line 135: ‘PXRD’ - maybe use this acronym to avoid confusion with single crystal XRD.
 - “XRD” has been changed to “PXRD” in the revised manuscript.

15. Line 160: ‘specie’.
 - “specie” has been revised to “species” in the manuscript.

16. Line 162: How does this compare to the theoretical Pt mass loading?
 For Pt/C₆₀-1 = 21.3 wt% Pt

For Pt/C₆₀-2 = 11.9 wt% Pt

- The “1” and “2” in Pt/C₆₀-1 and Pt/C₆₀-2 refer to the charged molar ratio of C₆₀ to Pt(dba)₂ during the sample synthesis, rather than the proportion of C₆₀ in the synthesized samples. Thus, we can consider that the theoretical Pt mass loading of Pt/C₆₀-1 and Pt/C₆₀-2 are very similar, which was confirmed by ICP-OES results (26.56 wt % for Pt/C₆₀-1) and (21.54 wt% for Pt/C₆₀-2) as shown in the Supplementary Table1.
17. This strongly suggests that the C₆₀-Pt polymer was formed with 1:1 C₆₀:Pt, hence any excess C₆₀ was removed. Again, what was the purpose of using Pt/C₆₀-2 being the main material of study? (over Pt/C₆₀-1. Note, in practice, the quoted loadings make little sense because the catalyst is diluted with ketjen black to improve conductivity, hence loadings are lower in reality.
- Please see response to reviewer #2 point #13 on page #10.
18. Line 177: ‘large number’ is quite vague. Any attempt at quantisation would be useful (atoms per nm²?)
- Since the single atoms are not uniformly distributed and the sample is stacked on the TEM copper mesh, it is hard to accurately quantify the numbers of single atoms. Based on some clear areas, we estimate the average number of atoms in 1 nm² is 3-5.
19. Line 177: this is the biggest problem here. The STEM images here and in the SI clearly show several crystalline and amorphous Pt particles, which will contain the vast majority of Pt atoms.
- Yes, the catalyst is a mixture of Pt single atoms and Pt clusters. The structure of Pt/C₆₀-2 has been characterized by PXRD, FT-IR, Raman, XPS, SEM, TEM, HAADF-STEM, CV and XAS measurements, which are techniques that are widely used to characterize metal single atoms and metal clusters. Based on these measurements, Pt/C₆₀-2 consists of a mixture of Pt single atoms and small Pt clusters. The poisoning experiments with KSCN and EDTA (Fig. 6a) and XPS results (Fig. 6d) also support the presence of both Pt single atoms and Pt clusters. Although Pt/C₆₀-2 is a mixture of single atoms and clusters, single atoms are predominant, which was verified by cyclic voltammetry (CV) and X-ray absorption spectroscopy (XAS) measurements. CV curves of metal single-atom catalysts showed no features associated with H₂ adsorption-desorption due to the oxidation state of the metal species. As displayed in Supplementary Fig. 6 in the revised Supplementary information, unlike Pt/C, which is dominated by metallic Pt nanoparticles (see Supplementary Fig. 7 in the revised Supplementary information), the CV curve of Pt/C₆₀-2 exhibits no hydrogen adsorption-desorption characteristic peaks, indicating that Pt atoms in Pt/C₆₀-2 are primarily atomically dispersed. XAS data also provide evidence for the predominance of Pt single atoms. Compared with the high intensity peak corresponding to Pt-C coordination (Fig. 4b), the significantly smaller peak of Pt-Pt coordination indicates a small number of Pt clusters. Pt is mainly atomically dispersed in Pt/C₆₀-2, consistent with the TEM results and CV measurements. Based on the XAS measurements, approximately 75% of Pt is bonded to C in the Pt/C₆₀-2 sample, and 25% is metallic Pt⁰. Based on the XPS results, 87.8% is Pt²⁺ and 12.2% is Pt⁰.
 - More HAADF-STEM images were provided in the revised Supplementary information. HAADF-STEM images of Pt/C₆₀-2 over a wider area revealed the spherical morphology and the high-loading of Pt single atoms as well as some Pt clusters (Fig. R7). Although both Pt single

atoms and Pt clusters were detected in the sample, HAADF-STEM images taken in different regions indicated mainly the presence of atomically dispersed Pt (Fig. R8). The predominance of Pt single atoms was further confirmed by cyclic voltammetry (CV) and X-ray absorption spectroscopy (XAS) measurements as discussed above.

- The above comments have been added to the manuscript. Fig. R7 and Fig. R8 have been added as Supplementary Fig. 3 and Supplementary Fig. 5, respectively in the revised Supplementary information.

Fig. R7. (a-d) HAADF-STEM images of Pt/C₆₀-2. Stepwise observations of the HAADF-STEM images revealed mainly isolated Pt single atoms.

Fig. R8. (a-c) HAADF-STEM images of Pt/C₆₀-2 taken in different regions indicated mainly the presence of atomically dispersed Pt.

- In addition, the enlargement of some cluster-like areas indicated mainly atomically dispersed Pt species, although we cannot completely discard few bonding for Pt atoms (Fig. R9).

Fig. R9. (a-e) Enlarged images of some cluster-like areas.

20. Line 178: Much too high accuracy in measurement. At 200 kV the electron wavelength is about 3 pm, and the pixel size is about 25 pm, which will be the limiting factor in resolution. Quoting distance measurements to 1 pm is nonsense here. Quoting to two decimal places is more appropriate (0.28 nm) In the SI there are 4 d-spacing measurements made all of which are quoted as exactly 0.227 nm, which does not seem plausible, even if the software is quoting to 3 decimal places.

- Many thanks for the comments. We measured the spacing of the lattice stripes of Pt nanoparticles with a Digital Micrograph. For example, in Fig 3d, we first measured $4d \approx 0.91$ nm and then calculated the d value as $0.91/4 = 0.227$. We have revised 0.227 nm to 0.23 nm in all related figures of the revised manuscript.

21. Line 180: Are there? What is the proof that they are single atoms and not clusters? The expected morphology of the catalyst is a 1D chain. Are there areas that show chains of Pt atoms. What is the expected separation of these based on models and how close are these to reality?

- Please see response to reviewer #2 point #19 on page #11-13.
- As for the 1D chain structure, we did not see areas that show chains of Pt atoms. The statement of “one-dimensional chain” was quoted from the source papers (*Journal of the Chemical Society, Chemical Communications* 1992, (4), 377 and *Chemistry Letters*, 1994, 1207).
- Pt/C₆₀-2 has a platinum-C₆₀ polymeric structure, where both Pt single atoms and Pt clusters are anchored on C₆₀. For a single Pt atom interacting with two C₆₀, it is preferably at the bridge position between two hexagonal rings (η^2), forming 4 Pt-C bonds. Moreover, the η^2 coordination mode has been experimentally observed for C₆₀-Pt complexes. For a metal cluster interacting with two C₆₀, there are three types of bonding modes, in which the bonding across the hexagonal face of C₆₀ ($\mu_3\text{-}\eta^2 : \eta^2 : \eta^2\text{-C}_{60}$) is the most stable one. And this type of bonding mode has been recognized in C₆₀-metal cluster complexes. In view of the polymeric structure of Pt/C₆₀-2, the minimum periodic models for Pt single atoms (C₆₀-Pt-C₆₀) and Pt clusters (C₆₀-Pt₁₃-C₆₀) were established, respectively. In C₆₀-Pt-C₆₀, the Pt atom is bonded to four carbon atoms from two C₆₀ molecules (Fig. 7a). In C₆₀-Pt₁₃-C₆₀, there are two types of Pt atoms, one is connected to the C₆₀ cage, and the other is connected to adjacent Pt atoms (Fig. 7b). The above discussion has been added in the **Density functional theory calculations** section of the revised manuscript.

22. Line 182: What proportion of surface Pt atoms would this particle size have? How would this impact on the activity of the catalyst?

- Obtaining an accurate proportion of Pt atoms on the particle surface is experimentally very difficult. Thus, we build a model to estimate the proportion of surface Pt atoms, although it may not represent the real numbers of Pt atom. The crystal structure of Pt is assumed to be *fcc*, thus the model was constructed based on the (111) crystal surface with a diameter of 1.72 nm (Fig. R10). The (111) crystal surface is arranged in the order of ABCABC, so the number of Pt atoms in the cluster is calculated to be 147, with 92 surface Pt atoms and 55 internal Pt atoms. The proportion of the surface Pt atoms to the whole atoms in the model is calculated to be 62.58%, and the surface Pt atoms would contribute to the activity of the catalyst.

Fig. R10. Model for Pt cluster, showing the surface atoms in green.

23. Line 183: images look like STEM-EDX, if so please mention the correct technique. The choice of area and magnification for the EDX are odd, the images only show that C and Pt are both present fairly homogeneously. The high contrast areas in the HAADF probably correspond to Pt clusters, supported by STEM-EDX, further suggesting there are areas of agglomerated Pt. Additionally, what are the measured abundances of C and Pt from the EDX, and how to these compare to both experimental and theoretical Pt weight loading?

- Yes, Fig. S3 of the original submitted manuscript is the STEM-EDX images. In addition, as suggested, the choice of area and magnification for the EDX is not good to represent the structure of Pt/C₆₀-2. We have obtained better results on HAADF-STEM and the corresponding EDX mappings as shown in Figs. R11 and R12, thus Fig. S3 has been replaced by Figs. R11 and R12, which have been added as Supplementary Figs. 2 and 4 to the revised Supplementary information, respectively. The following comments “The HAADF-STEM image and the corresponding EDX mappings confirmed that Pt/C₆₀-2 is composed of Pt and C elements (Fig. R11d-f). Since the classical TEM analysis is challenging for single atoms and ultrasmall clusters, the specific form of the Pt component was further identified by atomic-resolution HAADF-STEM images. As expected, a large number of Pt single atoms were observed, together with some Pt clusters. The average size of Pt clusters is about 1.7 nm. Although both Pt single atoms and Pt clusters were detected in the sample, HAADF-STEM images taken in different regions indicated mainly the presence of atomically dispersed Pt (Fig. R8)” have been added in the revised manuscript. Fig. R11, Fig. R12, and Fig R8 have been added to the revised manuscript as Supplementary Fig. 2, Supplementary Fig. 4, and Supplementary Fig. 5, respectively.
- The Pt content of Pt/C₆₀-2 measured by STEM-EDX is 19.19 wt%, similar to the experimental (21.54 wt%) and theoretical (21.3 wt%) Pt weight loading.

Fig. R11. (a-c) TEM images of Pt/C₆₀-2 at different magnifications with some small clusters (black spots in c) observed. (d-f) A representative of HAADF STEM image and the corresponding EDX mapping of Pt/C₆₀-2, showing the presence of Pt and C elements.

Fig. R12. (a-c) HAADF-STEM images of Pt/C₆₀-2 taken at different domains, showing the presence of Pt single atoms and relatively less Pt clusters. (d-f) HAADF-STEM image and the corresponding EDX mapping of Pt/C₆₀-2.

24. Lines 186-187: I'm not suggesting that areas of high contrast are all Pt clusters and not clusters of single atoms, it may well be the case that is suggested in the text. However, these TEM images are not sufficient to support that claim, indeed I believe they evidence the opposite case that the number of clusters greatly outweighs the number of single atoms.

■ Please see response to reviewer #2 point #19 on page #11-13.

25. Fig3b: What does this image show? An amorphous particle edge, but otherwise not much useful information here. How were the TEM grids prepared? This would be useful to know in order to see what the background in this image is.

■ TEM grids were prepared by dropping sample ink onto the copper mesh coated with carbon film. To distinguish the sample and the carbon film on the copper mesh, an orange line was drawn to mark the boundary between the Pt/C₆₀-2 sample and the amorphous carbon film. Pt/C₆₀-2 shows a spherical-like morphology and exhibits amorphous features as displayed in the TEM images. Since classical TEM analysis is challenging for metal single atoms and ultrasmall clusters, the specific form of the Pt component is further identified by atomic-resolution HAADF-STEM images. Therefore, we just use Fig. 3b to show the amorphous feature of the sample. Fig. 3b is the edge of a sphere to represent the amorphous feature of the catalyst. More TEM images with a wide area were provided (Fig. R11). Fig. R11 has been added as Supplementary Fig. 2 of the revised Supplementary information.

■ In addition, we apologize that the scale bar in Fig. 3c is wrong, it has been corrected. We also adjusted the order of Fig. 3c and Fig. 3d. Fig. 3 has been updated with Fig. R13 in the revised manuscript.

Fig R11. (a-c) TEM images of Pt/C₆₀-2 take at different domains. (d-f) HAADF STEM image and the corresponding EDX mapping of Pt/C₆₀-2, showing the presence of Pt and C elements.

Fig. R13. The (a) SEM, (b) HRTEM, and (c-d) HAADF-STEM images of Pt/C₆₀-2.

26. Fig3: STEM images - as above, there are many particles and agglomerations of Pt atoms. How big are these and what is the expected proportion of surface atoms. Additionally, what is the total expected ratio of single atoms to clusters.

- Please see response to reviewer #2 point #22 on page #14.
- Based on XAS results, approximately 75% of the Pt is bonded to C, and 25% is the metallic Pt⁰. XPS results are consistent with the existence of both Pt single atoms and Pt clusters in the catalyst. Deconvolution of the XPS spectrum of Pt/C₆₀-2 showed that there are 87.8% Pt²⁺ and 12.2% metallic state Pt⁰ in Pt/C₆₀-2.

27. Line 207: Coordination number is measured to be ~ 3, why is there a discrepancy if the coordination is Pt-C₄? A mixture of products or another product entirely? Maybe discuss why this difference could be present.

- Based on the XPS, Raman, and XAS results (Fig. 2c-d and Fig. 4), Pt atoms are mainly in a divalent form (Pt²⁺) which prefers to be four-coordination, thus one Pt binds to two C₆₀ in an η²-C₆₀ π-type bonding mode, forming 4 Pt-C bonds (Fig. 4c). Since the coordination number (CN) for Pt single atom is 4, the measured CN of 3 indicates that Pt clusters also connect with C₆₀ to lower the CN as the Pt atoms connected to C₆₀ in Pt clusters coordinate only 2 carbon atoms of C₆₀ (inset of Fig. 4c). The discussion has been added to the revised manuscript.

28. Line 210: Again, STEM results need more analysis to fully support this claim, at the moment STEM images appear to claim the opposite.

- Please see response to reviewer #2 point #19 on page #11-13.

Electrochemistry section:

29. why there is no discussion of LSV? The catalyst exhibits no features associated with HER (Fig.S4). Why?
- The discussion of LSV was presented in lines 226-235 of the originally submitted manuscript. Please find the discussion in the first paragraph of the “**Electrocatalytic HER performance of Pt/C₆₀-2**” section in the revised manuscript.
 - Unlike metal nanoparticles/clusters, CV curves of metal single atoms catalysts show no features associated with H adsorption-desorption because single atoms are in an oxidation state rather than the metallic state. As displayed in the CV curve of Pt/C₆₀-2 (Supplementary Fig. 6 of the revised Supplementary information), unlike Pt/C, which is dominated by metallic Pt nanoparticles, the CV curve of Pt/C₆₀-2 exhibits no hydrogen adsorption-desorption characteristic peaks, indicating the atomically dispersed structure of the Pt single atoms in Pt/C₆₀-2.
30. Electrocatalyst ink preparation procedure (sonication of Pt-C₆₀ compounds with nafion and ketjen in water/IPA for several hours) is very aggressive and may completely change the nature and structure of the catalyst. There is no guarantee that single atoms survive this process.
- To evaluate the stability of the Pt/C₆₀-2 catalyst after ink preparation, structure characterizations of the ink sample were conducted. The ink was prepared as follows: 5mg of Pt/C₆₀-2 and 2.5 mg ketjen black (KB) with 60 μL of Nafion solution were dispersed in 1440 μL of a water/isopropanol mixed solvent (1:1 volume ratio) and sonicated for 2 hours to form a homogeneous ink (denoted as Pt/C₆₀-2/KB). In order to do the characterizations, the ink was dried under a vacuum at 60 °C overnight. The PXRD, IR, Raman, and XPS results of the ink (Pt/C₆₀-2/KB) are consistent with those for pristine Pt/C₆₀-2 (Fig. R14a-d). Moreover, the atomically dispersed Pt single atoms are observed in the ink sample (Fig. R14e-f), confirming that the Pt single atoms remained after ink preparation, and the structure of Pt/C₆₀-2 is preserved. Furthermore, the HAADF-STEM images of the ink sample after the 100 h long-term stability test displayed atomically dispersed Pt atoms (see Fig. R4e), further illustrating the robust structure of the Pt/C₆₀-2 catalyst. Fig. R14 has been added as Supplementary Fig. 15 of the revised Supplementary information.

Fig. R14. (a) PXRD, (b) IR, (c) Raman, and (d) XPS spectra of the ink sample (Pt/C₆₀-2/KB) and Pt/C₆₀-2. (e,f) HAADF-STEM images of ink.

31. Note that C₆₀ reacts readily with KOH forming fullerols, so the catalyst will undergo further chemical changes when immersed in the electrolyte, i.e. it will not be the same material as it is believed and modeled in the paper.

- The synthesis of fullerol has been reported in literature such as *Journal of the Chemical Society Chemical Communications*, 1993, 23(23):1784-1785, and *Progress in Solid State Chemistry* 2016, 44, 59e74. In general, the synthesis of C₆₀ fullerol requires a phase transfer catalyst tetrabutylammonium hydroxide (TBAH) and oxidant H₂O₂, otherwise, the reaction is difficult to proceed. To assess the tolerance of Pt/C₆₀-2 against 1 M KOH, we leave Pt/C₆₀-2 in 1 M KOH overnight under stirring. The IR absorption bands of the KOH-treated Pt/C₆₀-2 are almost the same as that of fresh Pt/C₆₀-2, and the absorption band of O-H was not observed for Pt/C₆₀-2 (Fig. R15), indicating the absence of fullerol and the excellent tolerance of Pt/C₆₀-2 to KOH. These comments have been added to the revised manuscript. Fig. R15 has been added as Supplementary Fig. 16 of the revised Supplementary information.

Fig. R15. IR spectra of KOH-treated Pt/C₆₀-2 and C₆₀(OH)₂₄.

32. Line 290: Again, why were these ratios chosen? Nomenclature of materials is inconsistent
- For the ratios chosen, please see response to reviewer #2 point #8 on page #8.
 - For the nomenclature of materials, please see response to reviewer #2 point #2 on page #7.
33. Lines 291-292: Choice of TEM images in Fig 6b are poor to illustrate this point. It is not very clear what areas correspond to Pt clusters, and indeed why they are clusters here but not in the STEM imaging in Figure 3. Higher magnification imaging over a wider area would be useful. The second image of Fig 6b is not illustrative of anything other than amorphous material, which could be a range of things. EDX analysis would be useful again, both STEM mapping and qualitative abundances of C and Pt.
- It needs to mention that Fig 6b is the HR-TEM images of the comparison samples Pt/C₆₀-1 and Pt/C₆₀-4, while Fig 3c-d are the HAADF-STEM images of Pt/C₆₀-2. Pt/C₆₀-1 and Pt/C₆₀-4 were synthesized to elucidate the reduction of Pt clusters by increasing the charged ratio of C₆₀ during sample preparation, and thereby demonstrating the dominant contribution of Pt single atoms to HER activity. Pt/C₆₀-1 and Pt/C₆₀-4 were synthesized under the same reaction conditions as for Pt/C₆₀-2 except with different amounts of C₆₀.
 - In order to clearly observe Pt clusters, we show the TEM images with a scale bar of 5 nm (Fig. R16b), where the Pt lattices (0.23 nm corresponds to the Pt (111) crystal plane) can be observed. TEM samples were prepared by dropping sample ink onto a copper mesh coated with carbon film. To distinguish the sample and the carbon film on the copper mesh, an orange line was drawn to mark the boundary between the sample and the carbon film. Fig. 6b has been updated as Fig. R16b. TEM images over a wider area are given in Fig. R5, showing more abundant Pt clusters in Pt/C₆₀-1. Comparing the two images in Fig. R16b, we could know that the aggregation of Pt is alleviated in Pt/C₆₀-4 by adding more C₆₀ during the reaction. Consistent results were obtained from Pt 4f XPS analyses, where the Pt⁰ fraction was gradually decreased from 28.1 % for Pt/C₆₀-1 to 12.2 % for Pt/C₆₀-2, and 8.9 % for Pt/C₆₀-4. Moreover, Fig. 6d has been updated with Fig. R16d to show the presence of Pt⁰ fractions. Fig. R5 has been added as Supplementary 12 of the revised Supplementary information.

Fig. R16. (a) Catalyst poisoning experiments for Pt/C₆₀-2/KB in 1 M KOH with the addition of 10 mM EDTA or KSCN. (b) The HR-TEM for the control samples Pt/C₆₀-1 and Pt/C₆₀-4. (c) HER polarization curves for Pt/C₆₀-1 and Pt/C₆₀-4. (d) High-resolution XPS Pt 4f spectrum for Pt/C₆₀-1, Pt/C₆₀-2, and Pt/C₆₀-4.

Fig. R5. TEM images of (a) Pt/C₆₀-1 and (b) Pt/C₆₀-4 over a wider area, showing more abundant Pt clusters in Pt/C₆₀-1.

34. Lines 314-316: What about the Pt atom in the centre of the cluster? Should this say there are two types of ‘surface’ Pt atoms?

- Thank you very much for the correction. “two types of Pt atoms” has been revised to “two types of surface Pt atoms” in the revised manuscript. Pt atoms in the center of the cluster are connect to other Pt.

35. Fig 7b: Pt₁₃ cluster is bonded across the hexagonal face of C₆₀? Why was this bonding mode chosen?

- For a metal cluster interacting with C₆₀, there are three types of bonding modes, as shown in Fig. R17, in which the bonding across the hexagonal face of C₆₀ ($\mu_3\text{-}\eta^2 : \eta^2 : \eta^2\text{-C}_{60}$) is the most stable one. And this type of bonding mode has been widely recognized in C₆₀-metal cluster complexes.

Fig. R17. Bonding modes between metal cluster and C₆₀.

36. Line 348: For single Pt atom catalysis, adsorption of H₂O and H both have positive deltaG. Is positive deltaG for H₂O adsorption considered optimal?

- The negative adsorption energy indicates that the catalyst has strong adsorption to water, which is conducive to the subsequent water dissociation and provides sufficient H for H₂ production. For alkaline HER, the smaller the H₂O adsorption energy is, the better the catalyst is.

37. Lines 356-358: From Fig. 7f it is difficult to qualitatively assess the area under each curve in the blue rectangle. To my eye, I would say that there is a decrease from red to blue, then an increase from blue to yellow. Quantifying the area under each curve may be useful to support the claim of the text.

- By calculating the area under each curve, the active electron densities of Pt-5d near the Fermi level (highlighted by blue rectangular areas) are 1.231, 1.285, and 2.655 for C₆₀-Pt-C₆₀-1, C₆₀-Pt₁₃-C₆₀-2, and C₆₀-Pt₁₃-C₆₀-3, respectively. Fig 7f has been updated with Fig. R18 in the revised manuscript.

Fig. R18. PDOS for Pt atoms for C_{60} -Pt- C_{60} -1, C_{60} -Pt₁₃- C_{60} -2 and C_{60} -Pt₁₃- C_{60} -3.

38. Line 374: are these definitely fullerides? What is evidence for C_{60} being anion?

■ Please see response to reviewer #2 point #11 on page #9-10.

39. Line 390: 1H NMR? - please mention specific nucleus.

■ The 1H NMR spectrum of $Pt(dba)_2$ was recorded in $CDCl_3$ at 600 MHz.

40. Line 399: Observation - this is the exact same description as from the original Nagashima paper.

■ Sorry for our carelessness. The description has been changed to “After the mixture was stirred at room temperature for 60 hours under nitrogen, black precipitates were obtained and collected by filtration. Then the precipitates were washed with toluene until the filtrate was colorless and transparent.” in the revised manuscript.

Materials characterization section:

41. EDX - how was this recorded? (described as 'TEM mapping' in this text).

■ STEM-EDX was also performed on a Tecnai G2 F30 field-emission microscope. We've added this to the revised manuscript.

42. Lines 464-466: Again, evidence may be that not all Pt sites are exposed. From the average particle size data from TEM, estimating how the surface:bulk Pt atomic ratio assuming there are no single atoms will give a lower bound on the activity. How does this value compare to other literature and other experimental results in this text? Maybe estimation of the single-atom : particle is possible via this method - I think the single biggest unanswered question here.

■ Yes, not all Pt sites are exposed. We measured the number of exposed active sites through the CO-stripping method (*ACS Energy Lett.* 2021, 6, 1175–1180) and calculated the turnover frequency (TOF) values. TOF, defined as the number of molecules (e.g., H_2) produced per site per second, is regarded as the only metric of intrinsic activity (*ACS Energy Lett.* 2021, 6,

1175–1180). To evaluate the real number of active sites and the turnover of frequencies (TOF) for Pt/C₆₀₋₂/KB and Pt/C, characterization of catalyst surface area via CO-stripping was performed. The CV baseline was first measured at a scan rate of 10 mV s⁻¹ in the N₂-saturated 1.0 M KOH electrolyte. Then a reduction potential (0.05 V vs. RHE) was applied to the working electrode, during which CO was bubbled into the electrolyte for 15 min to completely trigger CO adsorption on catalyst, followed by the passage of N₂ for 15 min to allow the escape of CO from the electrolyte. The adsorbed CO was then oxidized through the CV scans, showing a distinct oxidation peak compared to the baseline. The number of active sites exposed in the catalyst was determined by integrating the area of the oxidation peak (6.21326*10¹⁵ for Pt/C₆₀₋₂ and 5.95049*10¹⁵ for Pt/C). Then the TOFs were calculated by using the following equation: $TOF = j \cdot N_A / n \cdot F \cdot \Gamma$, where the j , N_A , n , and Γ are current density, Avogadro number (6.023×10²³), number of electrons transferred ($n = 2$), Faraday constant (96485 C), and the number of active sites, respectively. The TOF values for Pt/C₆₀₋₂/KB are higher than those for Pt/C in a wide range of overpotential. For example, at 50 mV, 100 mV, and 150 mV, the TOF values are 2.17 s⁻¹, 5.55 s⁻¹, and 11.2 s⁻¹, respectively for Pt/C₆₀₋₂, which are almost twice the Pt/C values, and are comparable or even superior to the values of the reported catalysts (Fig. R19), suggesting the high intrinsic activity of Pt/C₆₀₋₂.

- The relevant description has been added to the revised manuscript. Fig. R19 has been added as Fig. 5e in the revised manuscript. A new Table R1 was added to the revised Supplementary information as Supplementary Table 4.

Fig. R19. TOF values for Pt/C₆₀₋₂/KB, 20 wt% Pt/C, and other reported advanced electrocatalysts.

Table R1. Comparison of the TOF values of Pt/C₆₀₋₂ with Pt/C and other reported catalysts in 1.0 M KOH.

Electrocatalysts	Overpotential (mV)	TOF (H ₂ s ⁻¹)	Ref.
	50	2.17	
Pt/C ₆₀₋₂	100	5.55	
	150	11.2	
	50	1.37	This work
20 wt% Pt/C	100	2.95	
	150	5.04	
Pt@DG	100	6.74	J. Am. Chem. Soc. 144 , 2171-2178 (2022).
Pt-SAs/MoS ₂	50	1.02	Nat. Commun. 12 , 3021 (2021).
Pt-SAs/WS ₂	150	6.41	Nat. Commun. 12 , 3021 (2021).
Pt/LiCoO ₂	200	2.25	Angew. Chem. Int. Ed. 59 , 14533–14540 (2020).
Pt/PtTe _x	50	1.43	Appl. Catal. B: Environ. 299 , 120640 (2021)
Pt-PdO	100	1.42	ACS Sustain. Chem. & Eng. 10 , 3704-3715 (2022)
Ru SAs-Ni ₂ P	190	3	Nano Energy 80 , 105467 (2021)
E-Co SAs	100	0.48	Adv. Funct. Mater. 31 , 2100 (2021)
α-Mo ₂ C	250	2.5	J. Mater. Chem. A 3 , 8361-8368 (2015)
Ru@C ₂ N	50	1.66	Nat. Nanotech. 12 , 441-446 (2017)
Pt ₁ SAC-VNGNMAs	100	4.1	J. Mater. Chem. A 7 , 15575-15579 (2019)
RhPd-H NPs	60	0.33	ACS Nano 13 , 12987-12995 (2019)
Ir@CON	50	0.66	Adv. Mater. 30 , 1805606 (2018)
RuCo	50	1.5	Nat. Commun. 9 , 4958 (2018)
Ru _{NP} @RuN _x -OFC/NC	100	0.49	Appl. Catal. B: Environ. 307 , 121193 (2022)

43. Line 480: should read 'structures'?

■ "structure" has been corrected to "structures" in the revised manuscript.

44. Reference 42: This is the wrong year of publication for this journal. It should be 1994.

■ The year for Reference 42 is 1992 (Nagashima, H.; Nakaoka, A.; Saito, Y.; Kato, M.; Kawanishi, T.; Itoh, K., $C_{60}Pd_n$: the first organometallic polymer of buckminsterfullerene. *Journal of the Chemical Society, Chemical Communications* **1992**, (4), 377-379). But the year for Reference 41 is wrong, we have corrected it from 2006 to 1994. Notably, due to the addition and removal of some references, ref. 41 and 42 have been updated as refs. 38 and 39, respectively.

Reviewer #3 (Remarks to the Author):

Single-atomic platinum on fullerene C60 surfaces for accelerated alkaline hydrogen evolution by Zhang, et al. deals with the synthesis and characterization of polymeric PtC60 species and their application as catalyst for the HER. The paper is well planned and executed, and clearly written, giving key literature. The authors claims that the PtC60 species outperforms the benchmark 20 wt% Pt/C. For me this is a key point of the paper, and I think that the authors do not compare both catalysts with the exact Pt loading, if I well understood. If the electrode was prepared that way : 5mg of the catalyst and 2.5 mg ketjen black with 60 μ L of nafion solution ; for all catalyst, the authors should consider the Pt available for catalysis, as the catalysts are very different in nature. That is, PtC60 should be less loaded than that of 20%Pt/C, as I assume the Pt is in nanoparticle form for the later (more information about the 20%Pt/C should be given in the manuscript). Maybe a chemisorption of both catalysts should be given and the catalytic tests corrected to this data. If authors consider this point, and it is confirmed that PtC60 outperforms Pt over carbon, the paper should be worthy to publish to this journal.

Apart from the chemisorption to obtain the Pt available for catalysis, other points should be revised:

It is true that PtC60 species have been prepared in the past, among others, but poorly characterized, I appreciate the effort that has been done in this sense. Nenertheless, the authors claim: C60 engineered single-atomic metal catalysts and their catalyticperformances have never been reported; but they should cite: 10.1039/d0cy00540a; 10.1021/acscatal.6b01429; 10.1039/c9cy02025j

They also claim: Therefore, we can conclude that Pt atoms bind to C60 molecules to form a one-dimensional metal-fulleride polymer with the chemical formula $[C_{60}\text{-Pt}]_n$ ($n > 1$), in which Pt and C60 are alternately arranged.⁴²; I have difficulties to understand this conclusion based in the Pt content.

In the DFT calculations part the authors compare adsorption energies of H2O and H on the surface of the single-atom and Pt13 coordinated to fullerene with that of Pt(111). It is known that the adsorption energies can vary from flat surfaces to nanoparticles, i twill be more accurate to compare with an bare Pt13 as well.

- We highly appreciate your valuable comments and suggestions on this work. Please see below our point-by-point responses to your concerns.
1. Single-atomic platinum on fullerene C60 surfaces for accelerated alkaline hydrogen evolution by Zhang, et al. deals with the synthesis and characterization of polymeric PtC60 species and their application as catalyst for the HER. The paper is well planned and executed, and clearly written, giving key literature. The authors claims that the PtC60 species outperforms the benchmark 20 wt% Pt/C. For me this is a key point of the paper, and I think that the authors do not compare both catalysts with the exact Pt loading, if I well understood. If the electrode was prepared that way : 5mg of the catalyst and 2.5 mg ketjen black with 60 μ L of nafion solution ; for all catalyst, the authors should consider the Pt available for catalysis, as the catalysts are very different in nature. That is, PtC60 should be less loaded than that of 20%Pt/C, as I assume the Pt is in nanoparticle form for the later (more information about the 20%Pt/C should be given in the manuscript). Maybe a chemisorption of both catalysts should be given and the catalytic tests corrected to this data. If authors consider this point, and it is confirmed that PtC60 outperforms Pt over carbon, the paper should be worthy to publish to this journal.
- Yes, to evaluate HER activity of a catalyst, the Pt available for catalysis should be considered. To obtain the number of Pt available for catalysis and the turnover frequency (TOF) values for both Pt/C₆₀-2 and Pt/C, CO-stripping experiments were performed. Please see response to reviewer #2 point #42 on page #23-25.
 - Besides TOFs, the mass activity (j_{mass}) can be used as a stand-in metric for activity. Thus we calculated the mass activity for both Pt/C₆₀-2 and Pt/C according to the equation: $j_{\text{mass}} = j/m \cdot w$, where the j , m , and w are current, mass loading of catalyst, and Pt weight content in catalyst, respectively. The results show that Pt/C₆₀-2 exhibits higher mass activity. Fig. R20 has been added as Supplementary Fig. 8c.

Fig. R20. Mass activity for Pt/C₆₀-2/KB and 20 wt% Pt/C in 1 M KOH.

- The structural characterizations of the commercial 20 % Pt/C were performed (Fig. R21). TEM and XRD results confirm the Pt nanoparticles (NPs) structures. Pt NPs with an average size of 2.63 nm are homogeneously distributed over the carbon matrix. Raman spectrum of Pt/C shows

the typical D-band and G-band of carbon. Fig. R21 has been added as Supplementary Fig. 7 in the revised Supplementary information.

Fig. R21. (a) TEM image, (b) PXRD, (c) XPS, and (d) Raman spectra of Pt/C.

Apart from the chemisorption to obtain the Pt available for catalysis, other points should be revised:

2. It is true that PtC₆₀ species have been prepared in the past, among others, but poorly characterized, I appreciate the effort that has been done in this sense. Nevertheless, the authors claim: C₆₀ engineered single-atomic metal catalysts and their catalytic performances have never been reported; but they should cite: 10.1039/d0cy00540a; 10.1021/acscatal.6b01429; 10.1039/c9cy02025j
 - These three instructive works have been described as “Metal-C₆₀ polymer nanocatalysts have been successfully synthesized and showed high catalytic performance for the reduction of quinoline and nitrobenzene. However, the application of C₆₀ engineered single-atomic metal catalysts in the field of electrocatalysis and their experimental catalytic performances have never been explored”. The three excellent works have been cited as references 35-37.
3. They also claim: Therefore, we can conclude that Pt atoms bind to C₆₀ molecules to form a one-dimensional metal-fulleride polymer with the chemical formula [C₆₀-Pt]_n (n > 1), in which Pt and C₆₀ are alternately arranged.⁴²; I have difficulties to understand this conclusion based in the Pt content.
 - We're sorry that we did not describe the structure clearly. The statement of “one-dimensional polymeric system of alternating C₆₀ and Pt atom” was quoted from the source papers (*Journal of the Chemical Society, Chemical Communications* 1992, (4), 377 and *Chemistry Letters*, 1994, 1207). However, according to current structure characterizations for Pt/C₆₀, in addition to Pt

single atoms, there are also Pt clusters. To correctly describe the structure, we have revised the description of the structure as “Therefore, we can speculate that Pt/C₆₀-2 is a high-loading Pt catalyst whose inner structure can be described as a platinum-C₆₀ polymeric structure, in which the predominant Pt single atoms serve as bridges connecting the adjacent C₆₀, and Pt clusters are anchored on C₆₀ as well”.

4. In the DFT calculations part the authors compare adsorption energies of H₂O and H on the surface of the single-atom and Pt₁₃ coordinated to fullerene with that of Pt(111). It is known that the adsorption energies can vary from flat surfaces to nanoparticles, it will be more accurate to compare with an bare Pt₁₃ as well.

■ We are grateful to the reviewer for making this proposal. The adsorption of H₂O and H on the surface of the Pt₁₃ cluster was calculated. As shown in the figure below, although the adsorption of H₂O on the Pt₁₃ cluster occurs spontaneously, the adsorption of *H needs to overcome a high energy barrier of 1.35 eV, which greatly impedes efficient H₂ formation. But when Pt₁₃ was connected to C₆₀, the adsorption of *H was remarkably enhanced owing to the electron-withdrawing property of C₆₀. These comments have been added to the revised manuscript, and Fig. 7e has been updated with Fig. R22 in the revised manuscript.

Fig. R22. The schematic diagram of the adsorption of *H₂O and *H on the surface for C₆₀-Pt-C₆₀-1, C₆₀-Pt₁₃-C₆₀-2, C₆₀-Pt₁₃-C₆₀-3, Pt (111), and Pt₁₃ cluster.

Reviewer #4 (Remarks to the Author):

The manuscript by Zhang et al. discusses a novel organometallic Pt(C₆₀) oligomer catalyst for H₂ evolution reaction. In the first part of the manuscript they describe the synthesis of the material and carefully characterize it using a variety of X-ray techniques (XPS, XANES), IR and Raman spectroscopy and microscopy. Based on this analysis they conclude that they successfully synthesized the target complex. This analysis is very detailed and the used techniques definitely support this conclusion. The obtained catalyst is then characterized by electrochemical techniques

(voltammetry and impedance spectroscopy) based on which they conclude that the obtained material is slightly more active than pure Pt(111). This is then followed by an analysis of the potential active site using catalysts with decreasing amount of Pt nanoparticles and through poisoning experiments. Here the authors convincingly show that the Pt(C₆₀)₂ complex is indeed the active species for H₂ evolution. These results are then rationalized by DFT modelling to obtain a detailed understanding of the underlying mechanisms. This section is unfortunately not fully convincing.

Overall this is a very careful and interesting study which has been summarized in a well written manuscript. I can support its publication after a few minor questions have been clarified.

Minor issues:

- We highly appreciate your valuable comments and suggestions on this work. Please see below our point-by-point responses to your concerns.
1. p.2: In the introduction the authors claim that already today electrolysis stands out for its simplicity and the availability of catalytic materials. This is unfortunately not yet true. The HER requires scarce and expensive Pt as a catalyst which can not yet be replaced easily. In line with this, the main technique to obtain H₂ in industry is steam reforming of natural gas.
 - The relevant statement has been changed to “Among the current hydrogen production methods, electrocatalytic hydrogen evolution reaction (HER) is a promising technique for its simplicity and environmental friendliness”.
 2. p.2: In the introduction the authors claim that the HER kinetics are hindered by oxygen intermediates. This sounds a bit strange. I would expect, that water reduction results in the formation of adsorbed *H and a dissolved OH⁻ not the formation of adsorbed *OH or *O groups. It would be good if this issue is clarified by the authors.
 - The statement of “chemisorption of the hydrogen and oxygen intermediates” has been revised to “adsorption of *H and desorption of H₂” in the revised manuscript.
 3. line 249: Impedance spectroscopy indicates a lower charge transfer resistance for the Pt(C₆₀) complexes. Why do the authors believe that this is not the sole reason for the very minor improvement in the performance?
 - Catalysts with small charge transfer resistance can lower the resistance between active sites and supports, and facilitate the transportation of electrons, resulting in higher HER activity. The smaller charge transfer resistance of Pt/C₆₀₋₂/KB is indeed one of the reasons for the better HER performance, but we believe it is not the only reason. Electrochemical measurements demonstrated that Pt/C₆₀₋₂ has a higher Electrochemical Active Surface Area (ECSA) than Pt/C, indicating that there are more accessible active sites in Pt/C₆₀₋₂ (the number of Pt sites involved in HER is 5.95049×10^{15} for Pt/C and 6.21326×10^{15} for Pt/C₆₀₋₂), which account for the better performance. Moreover, the number of H₂ molecules produced per Pt site per second (TOF) for Pt/C₆₀₋₂ is higher than that of Pt/C in a wide range of overpotential, suggesting the higher intrinsic activity of Pt/C₆₀₋₂. In addition to activity, stability is another important factor to evaluate the catalyst performance, Pt/C₆₀₋₂ with Pt species stabilized by C₆₀ exhibited much better durability than Pt/C (see Fig R6). Moreover, density functional theory calculations

demonstrated that Pt/C₆₀-2 has more appropriate H₂O and H adsorption energy than Pt (111), which are essential for effective alkaline HER.

4. line 294: The authors state here that the Pt/C₆₀-2 catalyst solely contains Pt(C₆₀). This is inconsistent with earlier parts of the manuscript where they show that at least minor amounts of Pt nanoparticles are present.
 - About the structure description, please see response to review #2 point #19 page #11-13. Based on structure characterizations of Pt/C₆₀ catalysts, as well as the poisoning experiments, Pt/C₆₀ catalysts are high-loading Pt catalysts with a mixture of Pt single atoms and Pt clusters. XPS results are consistent with the existence of both Pt single atoms and Pt clusters in the catalysts (Fig. R16d), as evident from the STEM images. Fig. 6b,d has been updated with Fig. R14b,d.

Fig. R16. (a) Catalyst poisoning experiments for Pt/C₆₀-2/KB in 1 M KOH with the addition of 10 mM EDTA or KSCN. (b) The HR-TEM for the control samples Pt/C₆₀-1 and Pt/C₆₀-4. (c) HER polarization curves for control samples Pt/C₆₀-1 and Pt/C₆₀-4. (d) High-resolution XPS Pt 4f spectrum for Pt/C₆₀-1, Pt/C₆₀-2, and Pt/C₆₀-4.

5. line 311: The model description is not completely clear. Do the authors use a periodic model of the Pt(C₆₀) complex or does it consist of a single isolated Pt(C₆₀)₂ monomer? If the later is true why did they choose a plane wave code with pbc rather than one of the standard molecular codes which would give access to better DFT functionals and more advanced solvation models? For

the periodic model also the box size and the amount of vacuum between the layers must be described in the computational details.

- Based on our experimental characterizations, Pt/C₆₀-2 is a high-loading Pt catalyst whose inner structure can be described as a platinum-C₆₀ polymeric structure, in which the predominant Pt single atoms serve as bridges connecting the adjacent C₆₀, and Pt clusters are anchored on C₆₀ as well. For a single Pt atom interacting with two C₆₀, it is preferably at the bridge position between two hexagonal rings (η^2), forming 4 Pt-C bonds. Moreover, the η^2 coordination mode has been experimentally observed for C₆₀-Pt complexes (*Angewandte Chemie International Edition* **1998**, 37 (13-14), 1916-1919; *Science* **1991**, 252 (5009), 1160-1161). For a metal cluster interacting with two C₆₀, there are three types of bonding modes (*Accounts of Chemical Research* **2003**, (1), 36), of which the bonding across the hexagonal face of C₆₀ ($\mu_3\text{-}\eta^2 : \eta^2 : \eta^2\text{-C}_{60}$) is the most stable mode (*ACS Catalysis* **2016**, 6 (9), 6018-6024; *Catalysis Science & Technology* **2019**, 9 (24), 6884-6898.). And this type of bonding mode has been widely recognized in C₆₀-metal cluster complexes (*Coordination Chemistry Reviews* **2016**, 308, 236-345; *Chemical Reviews* **2016**, 116 (6), 3812-3882). In view of the polymeric structure of Pt/C₆₀-2, we established the minimum periodic models for Pt single atom (C₆₀-Pt-C₆₀) and Pt clusters (C₆₀-Pt₁₃-C₆₀) to perform the calculations. These comments and related references have been added to the manuscript.
 - The cell parameters for Pt₁₃, C₆₀-Pt-C₆₀, and C₆₀-Pt₁₃-C₆₀ structures were set as a = 12 Å, b = 25 Å, and c = 12 Å. The Monkhorst-Pack Γ -centered grids of 1×1×1 for all structures. A vacuum space of 15 Å was employed to eliminate the interaction between the neighboring layers. These comments have been added to the manuscript.
6. Figure 7: The mechanism suggested by the authors which proceeds through the adsorption of H₂O followed by the formation of an H adsorbate is not fully convincing. They argue that owing to a thermodynamic barrier for H₂O adsorption Pt(C₆₀) is a better catalyst than Pt(111). On the other hand the authors show, that the adsorption energy of *H (which is used as descriptor) is with -160 mV for Pt(111) much closer to the ideal than any of the Pt(C₆₀) complexes. Thus, I would expect that Pt(111) is the best catalyst. Disregarding Pt(111) on the other hand based on the H₂O adsorption energy is in my opinion not fully correct since H adsorption could also occur through an Eley-Rideal type mechanism.
- It is commonly accepted that the HER activity in the acid media is entirely determined by hydrogen bonding energy. While the activity of alkaline HER is determined by both water adsorption and H adsorption. The alkaline HER mechanism is typically divided into the water adsorption/dissociation process ($\text{H}_2\text{O} + \text{e}^- \rightarrow \text{OH}^- + \text{H}_{\text{ad}}$, Volmer step, Tafel slope = 120 mV dec⁻¹) and the desorption of H₂ ($\text{H}_2\text{O} + \text{e}^- + \text{H}_{\text{ad}} \rightarrow \text{H}_2 + \text{OH}^-$, Heyrovsky step, Tafel slope = 40 mV dec⁻¹ or $\text{H}_{\text{ad}} + \text{H}_{\text{ad}} \rightarrow \text{H}_2$, Tafel step, Tafel slope = 30 mV dec⁻¹). The calculated Tafel slope of Pt/C₆₀-2 is 55 mV dec⁻¹, suggesting a Volmer-Heyrovsky process with the rate-determining step (RDS) of the Heyrovsky step. In other words, water adsorption is a prerequisite for providing sufficient H_{ad} for H₂ generation. The strong adsorption to H₂O will facilitate the H₂O capture rate of Pt sites to provide high concentration of H for hydrogen evolution. Highly efficient water adsorption on the surface of electrocatalysts is a crucial step for alkaline electrolytic water splitting. Based on the experimental and theoretical results, we believe that the synergistic effect leads to the better performance of Pt/C₆₀.

7. Figure 7: The difference in the H-adsorption energy between Pt(111) and Pt-C60-1 is with 140 mV rather minor. How large is the error the authors expect for their organometallic catalyst and how robust do they expect the trends to be with respect to the choice of functional/method (e.g. a meta-GGA or GGA+U)?

■ According to the reviewer's suggestions, we calculated the H-adsorption energy on Pt(111) and C₆₀-Pt-C₆₀-1 using the meta-GGA functional (Fig. R23). However, the results showed that meta-GGA does not perform well on HER since H-adsorption on Pt(111) is too bad. RPBE is the functional most recommended by Jens K. Nørskov (who has led the development of theory and the application of computational methods to study catalysis) for theoretic study of electrocatalysis, and it has been extensively used in electrocatalysis studies.

Fig. R23. Calculated adsorption energies of H on the surface for C₆₀-Pt-C₆₀-1 and Pt (111).

8. VIII) Figure 7: The adsorption energy of water on C₆₀-Pt₁₃ is with -1.43 eV extremely negative. This is somewhat surprising since I would expect that water only physisorbes extremely weakly. Do the authors have any explanation for this surprising result?

■ For the high adsorption energy of water on C₆₀-Pt₁₃-C₆₀, we would like to give the following explanations. Due to the electron-withdrawing properties of C₆₀, the electron redistribution between C₆₀ and Pt₁₃ would result in the positive Pt₁₃. The oxygen atom of the water molecule has a negative charge, so the positively charged Pt₁₃ is conducive to the adsorption of water due to its enhanced interaction with the oxygen atom in water. As a consequence, C₆₀-Pt₁₃-C₆₀ displays a much lower adsorption free energy for water. We have seen in published articles that the adsorption energy of water on the catalyst surface can be up to -1.45 eV (*Nat. Nanotechnol.* 2017, 12, 441e446), and -1.14 eV (*Adv. Mater.* 2020, 2000385).

REVIEWERS' COMMENTS

Reviewer #1 (Remarks to the Author):

The authors sufficiently revised their work according to what was proposed by me and I suggest acceptance of this contribution for publication.

Reviewer #3 (Remarks to the Author):

Review of revised manuscript NCOMMS-22-30291A: After a careful lecture of the rebuttal letter and the manuscript, I think that the authors have answered clearly and conveniently all the points raised by the reviewers. The manuscript is a very nice piece of work that in my opinion should be accepted for publication.

Reviewer #4 (Remarks to the Author):

I am fully satisfied with the answers provided by the referees and therefore recommend to publish the paper without any further changes.

April 6, 2023

Dear Reviewers,

We would like to thank all the reviewers for your time in reviewing our revised manuscript entitled “Single-atomic platinum on fullerene C₆₀ surfaces for accelerated alkaline hydrogen evolution” (Manuscript ID: NCOMMS-22-30291A). We greatly appreciate your valuable and constructive comments, which have greatly improved the manuscript.

With best regards,

Fang-Fang Li, Ph.D.

Email: ffli@hust.edu.cn

Response to Reviewers

REVIEWERS' COMMENTS

Reviewer #1 (Remarks to the Author):

The authors sufficiently revised their work according to what was proposed by me and I suggest acceptance of this contribution for publication.

■ Many thanks for the positive comments.

Reviewer #3 (Remarks to the Author):

Review of revised manuscript NCOMMS-22-30291A: After a careful lecture of the rebuttal letter and the manuscript, I think that the authors have answered clearly and conveniently all the points raised by the reviewers. The manuscript is a very nice piece of work that in my opinion should be accepted for publication.

■ Many thanks for the positive comments.

Reviewer #4 (Remarks to the Author):

I am fully satisfied with the answers provided by the referees and therefore recommend to publish the paper without any further changes.

■ Many thanks for the positive comments.